# Evolutionary Neural Architecture Search for Transformer in Knowledge Tracing

**Shangshang Yang**[1,3]    **Xiaoshan Yu**[1,3]    **Ye Tian**[1,3]    **Xueming Yan**[2]    **Haiping Ma**[1,3,4*]

**Xingyi Zhang**[1,3*]

[1]Anhui University    [2]Guangdong University of Foreign Studies
[3]Key Laboratory of Intelligent Computing and Signal Processing of Ministry of Education
[4]Department of Information Materials and Intelligent Sensing Laboratory of Anhui Province
`{yangshang0308, yxsleo, field910921}@gmail.com` `xueming126@126.com`
`hpma@ahu.edu.cn` `xyzhanghust@gmail.com`

## Abstract

Transformer has achieved excellent performance in the knowledge tracing (KT) task, but they are criticized for the manually selected input features for fusion and the defect of single global context modelling to directly capture students' forgetting behavior in KT, when the related records are distant from the current record in terms of time. To address the issues, this paper first considers adding convolution operations to the Transformer to enhance its local context modelling ability used for students' forgetting behavior, then proposes an evolutionary neural architecture search approach to automate the input feature selection and automatically determine where to apply which operation for achieving the balancing of the local/global context modelling. In the search space design, the original global path containing the attention module in Transformer is replaced with the sum of a global path and a local path that could contain different convolutions, and the selection of input features is also considered. To search the best architecture, we employ an effective evolutionary algorithm to explore the search space and also suggest a search space reduction strategy to accelerate the convergence of the algorithm. Experimental results on the two largest and most challenging education datasets demonstrate the effectiveness of the architecture found by the proposed approach.

## 1    Introduction

With the rapid development of online education systems, such as Coursera and ASSISTment [10], there have been a massive amount of available data about student-system interactions recorded in these systems. Based on the students' historical learning activities over time, knowledge tracing (KT), as a fundamental task in intelligent education, aims to trace students' knowledge states on knowledge concepts, which plays an important role in both personalized learning and adaptive learning [28, 12] and thus has received growing attention from the scientific community.

Since the concept of KT was first proposed by Corbett and Anderson in Bayesian knowledge tracing (BKT) [9], considerable efforts have been devoted to developing various KT approaches, which can mainly be split into two categories. The first type of KT approaches are traditional approaches based on probabilistic models or logistic models, such as dynamic BKT [17], LFA [3], PFA [30], and KTM [42]. The second type of approaches [21] have shown significantly better performance and generalization since they utilize various deep neural networks (DNNs) to solve the KT task, such as

---

*Corresponding authors.

recurrent neural networks (RNNs) as adopted in [53], long short-term memory networks (LSTMs) as adopted in [1], graph neural networks as adopted in [26], and Transformer as adopted in [7, 33, 36].

Among these DNN-based approaches, Transformer-based KT approaches have exhibited significantly better performance due to the better ability to tackle sequence tasks. However, as demonstrated in [16, 36], when there is no extra specific information introduced, the attention mechanism in the vanilla Transformer [41] cannot capture students' forgetting behavior [24], since its providing single global context modelling usually assigns high importance to the highly related records to predict the current record even with position information. However, those related records distant in time commonly pose little influence on the current prediction. Moreover, the input features in these approaches are manually selected based on domain knowledge and simply fused as the inputs to the Transformer encoder and decoder, but the manually selected features and the simple feature fusion method may miss some valuable features and complex feature interaction information.

To address the issues for further improving the performance, we first equip the Transformer with convolutions to boost its local context modelling ability used for students' forgetting behavior, then propose an evolutionary neural architecture search (NAS) [54] approach for KT (termed ENAS-KT), which aims to automatically select input features and determine where and which operation is applied to strike the balancing of local/global context modelling. Our main contributions are as:

- To the best of our knowledge, we are the first to apply NAS to search the Transformer architecture for the KT task. Here, the proposed ENAS-KT aims to find the best architecture that holds the optimal selected input features and the ideal balancing of the local/global context modelling, which can model students' forgetting behavior well and thus hold significantly good prediction performance on KT.

- In the proposed ENAS-KT, we design a novel search space for the Transformer and devise an effective evolutionary algorithm (EA) as the search approach to explore the search space. The search space not only considers the input feature selection of the Transformer but also equips it with local context modelling by replacing the original global path containing the attention module with the sum of a global path and a local path that could contain different convolutions. Besides, an effective hierarchical fusion method is suggested to fuse selected features. The proposed ENAS-KT not only employs a trained super-net for fast solution evaluation but also suggests a search space reduction strategy to accelerate the convergence.

- Experimental results validated on two largest and most challenging datasets show that the architecture found by the proposed approach holds significantly better performance than state-of-the-art (SOTA) KT approaches.

## 2 Related Work

### 2.1 Transformer-based Knowledge Tracing

Existing Transformer-based KT approaches focus on taking different features as the inputs to the Transformer encoder and decoder based on human expertise without changing the model architecture. In [7], Choi *et al.* proposed to directly employ the vanilla Transformer for KT, namely SAINT, where the exercise embedding and knowledge concept embedding are taken as the input of the Transformer encoder part and the response embedding is taken as the input to the Transformer decoder part. As a result, the SAINT achieves significantly better performance than other KT approaches. To model the forgetting behavior in students' learning, Pu *et al.* [33] further fed the timestamp embedding of each response record to the Transformer model. Besides, to model the structure of interactions, the exercise to be answered at the next time is also fed to the model. As the successor of the SAINT, due to the problem in SAINT that students' forgetting behavior cannot be modelled well by the global context provided by the attention mechanism, the SAINT+ [36] aims to further improve the performance of KT by mitigating the problem. To this end, two temporal feature embedding, the elapsed time and the lag time, are combined with response embedding as input to the decoder part.

Different from above two approaches that only focus on how to manually select input features to alleviate the forgetting behavior problem, this paper focuses on not only realizing the input feature automatic selection of the Transformer but also the Transformer architecture automatic design by proposing an EA-based NAS approach, where convolution operations are considered in the Transformer to enhance its local context ability used for students' forgetting behavior.

## 2.2 Neural Architecture Search for Transformer

NAS has been applied to many domains [13], such as convolution neural networks in computer vision [54], RNNs in natural language processing [46] or automatic speech recognition [2].

So *et al.* [38] are the first to apply NAS to design the Transformer architecture, where an EA is employed to explore the devised NASNet-like search space consisting of two stackable cells for translation tasks. To save computation resources, a progressive dynamic hurdles method is employed to early stop the training of poor performance architectures. Inspired by this work, there have been a series of NAS work proposed to search the Transformer architecture in different research fields. For NLP, the representative approaches include NAS-BERT [49] and HAT [45]. For ASR, the representative approaches include the LightSpeech [23], and DARTS-CONFORMER [35]. For CV, the representative approaches include the AutoFormer [6], GLiT [5], and ViTAS [39]. The Transformer architectures found by these approaches have demonstrated significantly better performance than the SOTA competitors in their own domains.

Despite the emergence of these NAS approaches for the Transformer, their designed search spaces for different domains cannot be directly used for solving the feature selection problem and the problem of balancing local context and global context in this paper. To the best of our knowledge, we are the first to apply the NAS to search the Transformer architecture for the KT task. Different from NAS-Cell [11] that directly applies existing NAS methods used for general LSTMs to KT, our work designs effective search space for the Transformer to strengthen its modeling capability in KT.

# 3 Preliminaries

## 3.1 Knowledge Tracing Problem

Given a student's response records on a set of exercises over time, i.e., a sequence of interactions denoted by $I = \{I_1, I_2, \cdots I_n\}$, the KT task [21] aims to predict the probability that student answers correctly on the exercise in the next interaction $I_{n+1}$. $I_i = (e_i, o_i, dt_i, r_i)$ is the $i$-th interaction of the student, $e_i$ denotes the the $i$-th exercise assigned to the student, and $o_i$ contains other related information about the interaction $I_i$, such as the timestamp, the type of exercise $e_i$, knowledge concepts in $e_i$, and so on. $dt_i$ represents the elapsed time the student took to answer, $r_i \in \{0, 1\}$ is the student's response/answer on exercise $e_i$, where $r_i$=1 means the answer is correct otherwise the answer is incorrect.

Based on the interactions before the $t$+1-th interaction $I_{t+1}$, the KT focuses on predicting the probability of the student answering correctly on exercise $e_{t+1}$, formally:

$$P(r_{t+1}|I_1, I_2, \cdots, I_t, \{e_{t+1}, o_{t+1}\}),\tag{1}$$

where the response $r_{t+1}$ and the elapsed time $dt_{t+1}$ cannot be accessed for the prediction of $I_{t+1}$.

## 3.2 Transformer-based KT

Transformer [41] consists of an encoder part and a decoder part, where each part is composed of a stack of $N$ identical blocks, and each block has a multi-head self-attention (MHSA) module and a position-wise feed-forward network (FFN) module. Then, each encoder block can be expressed as

$$block_{En} : h = \text{LN}(\text{FFN}(h) + h), \ h = \text{LN}(\text{MHSA}(X, X, X) + X),\tag{2}$$

where $X$ is the input of the encoder block, and $\text{LN}(\cdot)$ is layer normalization [48]. Each decoder block can be denoted by

$$block_{De} \begin{cases} h = \text{LN}(\text{Masked\_MHSA}(X, X, X) + X) \\ h = \text{LN}(\text{FFN}(h) + h), \ h = \text{LN}(\text{MHSA}(h, O_{En}, O_{En}) + h) \end{cases},\tag{3}$$

where $\text{Masked\_MHSA}(\cdot)$ is the $\text{MHSA}(\cdot)$ with the mask operation to prevent current position to attend the later positions, and $O_{En}$ is the last encoder block's output.

For the Transformer-based KT approaches, the following sequential inputs $X_{input}$ with length of $L$:

$$X_{input} = \{E, O, DT, R\} = \{E = \{e_1, e_2, \cdots, e_L\},$$
$$O = \{o_1, o_2, \cdots, o_L\}, DT = \{dt_{start}, dt_1, \cdots, dt_{L-1}\}, \ R = \{r_{start}, r_1, \cdots, r_{L-1}\}\}\tag{4}$$

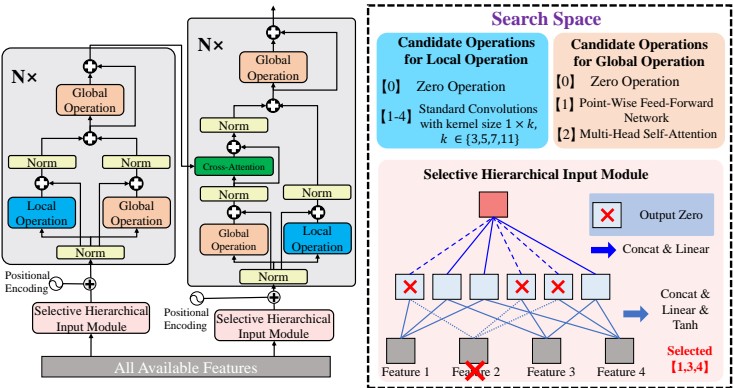

Figure 1: The proposed search space. There are five candidate operations for the local operation and three operations for the global operation.

will be first mapped into a set of embedding $X_{embed} = \{embed_i \in R^{L \times D} | 1 \leq i \leq Num\}$, where $D$ is the embedding dimension, $Num$ is the number of available features in a dataset, $dt_{start}$ and $r_{start}$ are the start tokens. $O$ could contain multiple features and thus produce multiple embedding. Then, two set of embedding $X_{En}$ and $X_{De}$ will be selected from $X_{embed}$ and fed to the encoder part and decoder part, respectively. Finally, the decoder will output the predicted response $\hat{r} = \{\hat{r}_1, \hat{r}_2, \cdots, \hat{r}_L\}$ as

$$\hat{r} = Decoder(Fusion(X_{De}), O_{En}, O_{En}), \quad O_{En} = Encoder(Fusion(X_{En})), \quad (5)$$

where $Fusion(\cdot)$ is a fusion method to get a $L \times D$ tensor. $Encoder(\cdot)$ and $Decoder(\cdot)$ denote the forward passes of the encoder part and decoder part, whose each block's forward pass is as shown in (2) or (3). Due to the nature of KT task, a mask operation is necessary for each MHSA module to prevent current position from accessing the information in later positions. Thus the MHSA module mentioned in subsequent sections refers to $Masked\_MHSA(\cdot)$.

Existing Transformer-based KT approaches empirically fuse the exercise-related embedding and response-related embedding as the inputs to encoder and decoder parts, respectively. However, the adopted feature fusion method is simple and the selected feature embedding sets may be not the best choice. Therefore, we further employ NAS to realize the input feature selections of both encoder and decoder parts and suggest an effective hierarchical feature fusion.

## 4 The Proposed ENAS-KT

To achieve the automatic input feature selection and automate the balancing of local/global modelling, we design a novel search space for the Transformer, and propose an evolutionary algorithm to search the optimal architecture. The following two subsections will present the details.

### 4.1 Search Space Design

We first replace post-LN [48] with the pre-LN [19] to stabilize the training of the Transformer. Then, we design a novel search space for the Transformer, which has been presented in Fig. 1 and there exist three important designs: (1) A selective hierarchical input module; (2) The sum of a global path and a local path to replace the original global path; (3) The global operations containing MHSA module, FFN module, and zero operation.

#### 4.1.1 Selective Hierarchical Input Module

Given all available input features $X_{input} = \{E, O, DT, R\}$ in a student's response records, we can obtain each candidate embedding in $X_{embed} = \{embed_i \in R^{L \times D} | 1 \leq i \leq Num\}$ by categorical embedding or continuous embedding. For example, the categorical embedding of $E \in R^{1 \times L}$ can be computed by

$$embed_{cate} = one\text{-}hot(E) * \mathbf{W}_{cate}, \mathbf{W}_{cate} \in R^{num_E \times D}, \quad (6)$$

where $num_E$ is the number of all exercises, $one\text{-}hot(\cdot)$ outputs a $L \times num_E$ one-hot-vector matrix, and $\mathbf{W}_{cate}$ denotes the trainable parameters of the embedding layer. And the continuous embedding of $DT \in R^{1 \times L}$ can be computed by

$$embed_{cont} = DT^T * \mathbf{W}_{cont}, \mathbf{W}_{cont} \in R^{1 \times D}. \tag{7}$$

Based on all candidate embedding $X_{embed}$, a selective hierarchical input module is proposed to first select two sets of suitable embedding from $X_{embed}$ and then fuse the selected embedding as the inputs to encoder and decoder, respectively. To this end, we employ two binary vectors to determine which candidate embedding is selected:

$$\mathbf{b_{En}} = (b_i \in \{0,1\})^{1 \times Num}, 1 \le sum(\mathbf{b_{En}}), \ \mathbf{b_{De}} = (b_i \in \{0,1\})^{1 \times Num}, 1 \le sum(\mathbf{b_{De}}), \tag{8}$$

where $1 \le i \le Num$ and $1 \le sum(\mathbf{b_{En}})$ ensures at least one embedding is selected. $b_i = 1$ means embedding $embed_i$ in $X_{embed}$ is selected and $b_i = 0$ otherwise. By doing so, two sets of selected embedding $X_{embed}^{En}$ and $X_{embed}^{De}$ used for the encoder and decoder parts can be obtained by

$$X_{En} = \{embed_i \in X_{embed} | b_i = 1, b_i \in \mathbf{b_{En}}\}, \ X_{De} = \{embed_i \in X_{embed} | b_i = 1, b_i \in \mathbf{b_{De}}\}, \tag{9}$$

Instead of directly applying the simple sum or concatenation operation to $X_{En}$, we propose a hierarchical fusion method to fully aggregate the selected embedding $X_{En}$ as

$$Hier(X_{En}): \ input_{En} = concat(temp) * \mathbf{W}_{out}, \ temp = \cup_{x_i, x_j \in X_{En}}^{x_i \ne x_j} \sigma(concat([x_i, x_j]) * \mathbf{W}_{ij}), \tag{10}$$

where $concat(\cdot)$ is concatenation operation, $\sigma(\cdot)$ is the tanh [14] activation function, $\mathbf{W}_{ij} \in R^{2D \times D}$ and $\mathbf{W}_{out} \in R^{|temp| * D \times D}$ are trainable matrices, and $|temp|$ denotes the number of temporary features in $temp$. The first item in equation (10) is to combine the selected embedding in pairs to fully fuse a pair of embedding without the influence of other embedding, which is inspired by the feature fusion in multi-modal tasks [25]. After obtaining $temp$, the second item in equation (10) is to directly apply the concatenation operation to $temp$ followed by a linear transformation to get the final input $input_{En}$ to the encoder part. The input $input_{De}$ to the decoder part can be obtained in the same way.

As shown in Fig. 1, the size of $\mathbf{W}_{out}$ is fixed as $\frac{Num*(Num-1)*D}{2} \times D$ for easily training super-Transformer, but the temporary features related to non-selected embedding are set to zeros. For example, feature 2 is not selected, thus the 1st, 4th, and 5th temporary features are set to zeros.

### 4.1.2 The Global Path and The Local Path

Since students' forgetting behaviors in the KT cannot be directly modelled well by the single MHSA module, we aim to incorporate the Transformer with the local context modelling. Thus, we split the original global path containing the MHSA module in each encoder block into the sum of two paths: one path is the original global path, and another path is the newly added local path containing convolution operations. Each decoder block is also set to be composed of these two paths. After applying the pre-LN to the input $X$ of each block as $x_1 = \text{LN}(X)$, each encoder and decoder block's forward pass in (2) and (3) can be rewritten as

$$block_{En} \begin{cases} gp = \text{LN}(x_1 + \text{MHSA}(x_1, x_1, x_1)) \\ lp = \text{LN}(x_1 + LO(x_1)) \\ h = h + \text{FFN}(h), h = lp + gp \end{cases} \text{ and } block_{De} \begin{cases} gp = \text{LN}(x_1 + \text{MHSA}(x_1, x_1, x_1)) \\ gp = gp + \text{MHSA}(gp, O_{En}, O_{En}) \\ lp = \text{LN}(x_1 + LO(x_1)) \\ h = h + \text{FFN}(h), h = lp + gp \end{cases}, \tag{11}$$

where $h$ refers to the output, $gp$ is the latent representation obtained by the global path, $lp$ is the latent representation obtained by the local path, and $LO(\cdot)$ denotes a local operation used for the local path.

As shown in Fig. 1, there are five candidate operations for $LO(\cdot)$, including four one-dimension convolution operations with kernel sizes in $\{3, 5, 7, 11\}$ and a zero operation that returns a zero tensor. When the zero operation is used for $LO(\cdot)$, the $block_{En}$ and $block_{De}$ will degenerate to the encoder and decoder block in the vanilla Transformer.

### 4.1.3 The Global Operation

As demonstrated in the Sandwich Transformer [32], reordering the MHSA model and FFN module may lead to better performance. For this aim, a global operation is used to replace the first MHSA

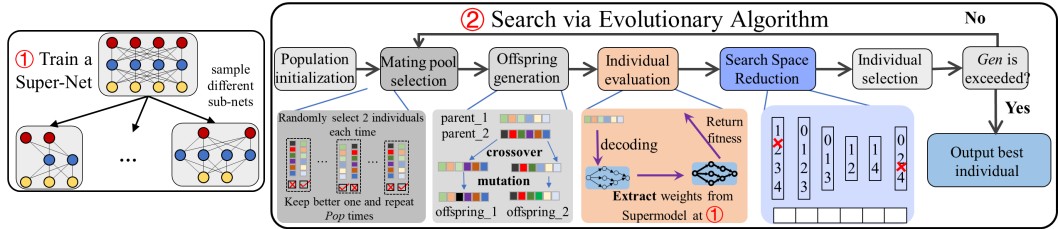

Figure 2: The overview of the proposed ENAS-KT.

module and the FFN module of each encoder and decoder block in (11), where $\text{MHSA}(\cdot)$ and $\text{FFN}(\cdot)$ are replaced by $GO_1(\cdot)$ and $GO_2(\cdot)$. Here, $GO_1(\cdot)$ and $GO_2(\cdot)$ represent the adopted global operation. As shown in the light orange area on the right of Fig. 1, there are three candidate operations for $GO_{1/2}(\cdot)$, including the MHSA module, FFN module, and a zero operation.

With the above modifications, the proposed Transformer architecture predicts the response $\hat{r} = \{\hat{r}_1, \hat{r}_2, \cdots, \hat{r}_L\}$ as

$$\hat{r} = Decoder(Hier(X_{De}), O_{En}, O_{En}), \ O_{En} = Encoder(Hier(X_{En})), \tag{12}$$

where $X_{En}$ and $X_{De}$ are obtained based on $\mathbf{b_{En}}$, $\mathbf{b_{De}}$, and $X_{embed}$ as shown in (8) and (9).

### 4.1.4 Encoding Scheme

Under the proposed search space, there theoretically exist $Num^2 * 2^{2(Num-1)} * (3*3*5)^{2N}$ candidate Transformer architectures in total, where each candidate Transformer architecture $\mathcal{A}$ can be encoded by $\mathcal{A} = [\mathbf{b_{En}}, \mathbf{b_{De}}, (lo \in \{0, \cdots, 4\}, go_1, go_2 \in \{0, 1, 2\})^{1 \times 2N}]$.

Here $\mathbf{b_{En}}$ and $\mathbf{b_{De}}$ denote which embedding is selected for the inputs to encoder and decoder parts, and $2N$ triplets $(lo, go_1, go_2)$ determine which operations are adopted for each block, where $lo$, $go_1$, and $go_2$ determine which operation is used for $LO(\cdot)$, $GO_1(\cdot)$, and $GO_2(\cdot)$.

Next, we search the optimal Transformer architecture by solving the following optimization problem:

$$\max_{\mathcal{A}} \mathbf{F}(\mathcal{A}) = f_{val\_auc}(\mathcal{A}, D_{val}), \tag{13}$$

where $f_{val\_auc}(\mathcal{A}, D_{val})$ denotes the AUC (*area under the curve*) value of $\mathcal{A}$ on dataset $D_{val}$.

## 4.2 Search Approach Design

**Overview.** An overview of the proposed approach is shown in Fig. 2, where a super-Transformer is first trained and then an EA with $Pop$ individuals and $Gen$ generations is used for searching.

### 4.2.1 Super-Transformer Training

Inspired by the sandwich rule [51], we propose to make the training rule adapt to our super-Transformer by utilizing three sub-models in one batch of training of the super-Transformer, including a global sub-model $sub_g$, a local sub-model $sub_l$, and a randomly sampled sub-model $sub_r$. $sub_g$ consists of pure global operations but no local operations, whose $\mathbf{b_{En}}$ and $\mathbf{b_{De}}$ are randomly generated; similarly, $sub_l$ consists of only convolution operations with the smallest kernel size.

### 4.2.2 Search Space Reduction-based Evolutionary Search

After training the supernet, a tailored EA is employed for searching by solving the problem in (13). The proposed EA takes the binary tournament selection [37] and single-point crossover together with the bit-wise mutation [40] to mate and generate individuals, where a search space reduction strategy is suggested to reduce the search space for accelerating the convergence at each generation.

**Search Space Reduction Strategy.** This strategy aims to iteratively remove some worse operations from the search space based on all individuals' fitness statistics to speed up the convergence. The item on each bit of $\mathcal{A}$ is an operation set, such as the item on first bit is $\{0, 1\}$, and the items on

Table 1: Statistics of the two largest education datasets for knowledge tracing, EdNet and RAIEd2020.

| Datasets/Statistics | # of interactions (# of tags) | # of students (# of tag-sets) | # of exercises (# of bundles) | # of skills (# of explanation) |
|---|---|---|---|---|
| **EdNet/RAIEd2020** | 95,293,926/99,271,300 (302)/(189) | 84,309/393,656 (1,792)/(1,520) | 13,169/13,523 (9,534)/(-) | 7/7 (-)/(2) |

Table 2: The comparison between ENAS-KT and compared approaches in term of model performance and parameters. The results under **splitting setting of 80% and 20%** on EdNet and RAIEd2020 are reported in terms of RMSE, ACC, and AUC, averaged over 5-fold cross-validation. The Wilcoxon rank sum test with a significance level $\alpha$=0.05 is used for analyzing the significance, where '+','−', and '≈' denote **Ours** is significantly better than, worse than, and similar to the compared approach, respectively. "M" is short for "million" in "Param.(M)", counting the number of model parameters.

| Dataset | Metric | DKT | HawkesKT | CT-NCM | SAKT | AKT | SAINT | SAINT+ | NAS-Cell | Ours |
|---|---|---|---|---|---|---|---|---|---|---|
| **Param.(M)** | EdNet | 0.13495 | **0.019578** | 1.9974 | 2.0864 | 1.2330 | 2.7492 | 3.1862 | 1.8692 | 3.8232 |
| | RAIEd2020 | 0.13531 | **0.019932** | 2.0431 | 2.1317 | 1.2335 | 2.7945 | 3.2315 | 1.9145 | 4.1262 |
| **EdNet** | **RMSE ↓** | 0.4653 | 0.4475 | 0.4364 | 0.4405 | 0.4399 | 0.4322 | 0.4285 | 0.4345 | **0.4209** |
| | **ACC ↑** | 0.6537 | 0.6888 | 0.7063 | 0.6998 | 0.7016 | 0.7132 | 0.7188 | 0.7143 | **0.7295** |
| | **AUC ↑** | 0.6952 | 0.7487 | 0.7743 | 0.7650 | 0.7686 | 0.7825 | 0.7916 | 0.7796 | **0.8062** |
| **RAIEd2020** | **RMSE ↓** | 0.4632 | 0.4453 | 0.4355 | 0.4381 | 0.4368 | 0.4310 | 0.4272 | 0.4309 | **0.4196** |
| | **ACC ↑** | 0.6622 | 0.6928 | 0.7079 | 0.7035 | 0.7076 | 0.7143 | 0.7192 | 0.7167 | **0.7313** |
| | **AUC ↑** | 0.7108 | 0.7525 | 0.7771 | 0.7693 | 0.7752 | 0.7862 | 0.7934 | 0.7839 | **0.8089** |
| +/-/≈ (six results totally) | | 6/0/0 | 6/0/0 | 6/0/0 | 6/0/0 | 6/0/0 | 6/0/0 | 6/0/0 | 6/0/0 | - |

the bits related to $lo$ are $\{0, 1, 2, 3, 4\}$. To start with, combine individuals in the population and offspring population; secondly, record each operation fitness in list $SF$, where each operation fitness is obtained by averaging the fitness of the individuals containing the corresponding operation; thirdly, compute fitness standard deviation of each each operation set as list $SF_{std}$; next, based on $SF$ and $SF_{std}$, the worst two operations will be deleted, and the worst operation in the operation set with the largest standard deviation will be also removed, where there is at least one operation kept for each bit. By doing so, the ever-reducing search space makes the search process easier. **More details** (about the super-net training, the EA, and the reduction strategy) can be found in Appendix A.

## 5 Experiments

### 5.1 Experimental Settings

**Datasets.** For convincing validation, two largest and most challenging real-world education datasets EdNet [8] and RAIEd2020 [34] were used in experiments, whose statistics are summarized in Table 1.

In addition to the above five feature embedding in Table 1: exercise $exer$, skill $sk$, tag $tag$, tag-set $tagset$, and bundle/explanation $bund/expla$, there are other seven feature embedding in candidate input embedding $X_{embed}$ ($Num = 12$), including answer embedding $ans$, continuous embedding for elapsed time $cont\_ela$, categorical embedding for elapsed time in terms of seconds $cate\_ela\_s$, continuous embedding for lag time $cont\_lag$, categorical embedding for lag time in terms of seconds, minutes, and days (termed $cate\_lag\_s$, $cate\_lag\_m$, and $cate\_lag\_d$).

**Compared Approaches**. The following eight SOTA KT approaches were used for comparison:

- LSTM-based and Hawkes-based methods: DKT [31], HawkesKT [44], CT-NCM [24], and NAS-Cell [11]. HawkesKT and CT-NCM consider extra students' forgetting behaviors in modelling, and NAS-Cell is NAS-based method for searching LSTMs' cells.

- Attention mechanism-based neural network methods: SAKT [27] and AKT [16]. Here AKT considers extra students' forgetting behavior in its attention weights.

- Transformer-based methods: SAINT [7] and SAINT+ [36]. But SAINT+ considers extra features of elapsed time and lag time in the decoder input to mitigate the forgetting behavior.

**Parameter Settings.** According to [7, 36], all students were randomly split into 70%/10%/20% for training/validation/testing. The maximal length of input sequences was set to 100 ($L$=100), we truncated student learning interactions longer than 100 to several sub-sequences, and 5-fold cross-validation was used. The number of blocks $N$=4, embedding size $D$=128, the hidden dimension of FFN was set to 128. To train the supernet, epoch number, learning rate, dropout rate, and batch size was set to 60, 1e-3, 0.1, and 128, and the number of warm-up steps in Noam scheme [7] was set to 8,000. To train the best architecture, epoch number and the number of warm-up steps was set to 30 and 4,000, and others kept same as above. $Pop$ and $Gen$ was set to 20 and 30. For a fair comparison, parameter settings of compared approaches are same as those in their original papers. We adopted AUC, *accuracy* (ACC), and *root mean squared error* (RMSE) as evaluation metrics. All experiments were implemented with PyTorch and run under NVIDIA 3080 GPU. The source code of the proposed approach is publicly available at `https://github.com/DevilYangS/ENAS-KT`.

## 5.2 Overall Results

To verify the effectiveness of the proposed approach, Table 2 summarizes the performance of the proposed ENAS-KT and all comparison KT approaches on predicting students' responses across EdNet and RAIEd2020 datasets, where the average RMSE, ACC, and AUC values over 5-fold cross-validation are reported and the best results are in bold. For more convincing, the Wilcoxon rank sum test [47] with a significance level $\alpha$=0.05 is used for analyzing the significance. In addition, the number of model parameters for each approach is reported for profound observations.

From Table 2, we can obtain the following five observations: (1) The performance of Transformer-based KT approaches is significantly better than LSTM-based KT approaches and attention mechanism-based KT approaches. (2) From the comparisons between DKT and CT-NCM, SAKT and AKT, SAINT and SAINT+, it can be found that considering the forgetting behavior in the KT can indeed improve the prediction performance to some extent under whichever type of neural architecture. (3) The RMSE, ACC, and AUC values achieved by the architecture obtained by the proposed approach on both two datasets are significantly better than that achieved by comparison approaches, which indicates the architecture found by the proposed approach is more effective in tracing students' knowledge states. (4) Compared to NAS-Cell, the architectures found by the proposed approach are more effective, which demonstrates the effectiveness of the proposed search space and search algorithm. (5) In terms of model parameters, HawkesKT is the best, DKT is the second-best, while the architectures found by the proposed ENAS-KT are worse than other KT models. Actually, high model parameters of **Ours** are attributed to its input part, which is used to get the input embeddings and aggregate these embeddings.

To analyze the influence of model parameters on the performance of the best-found architecture, four parameter constraints (including 2M, 2.5M, 3M, and 3.5M) are added to the proposed ENAS-KT, respectively, where the parameters of all generated architectures cannot exceed the used parameter constraint. As a result, four architectures are found on EdNet, denoted as ENAS-KT(2M), ENAS-KT(2.5M), ENAS-KT(3M), and ENAS-KT(3.5M), and Table 3 gives a brief comparison between these found architectures some KT models. As can be seen, the proposed ENAS-KT can still find good potential architecture under the parameter constraint, and the found architecture's performance increases with model parameters increasing, but the increasing in model parameters is caused by the change of the found model architecture, not the embedding or hidden size. Therefore, we can conclude that the effectiveness of the proposed approach is mainly not attributed to its high model parameters, to some extent, which can be mainly attributed to the suggested search space. Note that the comparison between SAINT/SAINT+ and SAINT(more)/SAINT+(more) can indirectly validate the above conclusion that only increasing the model parameters to improve model performance does not work and is unworthy, which further demonstrates the effectiveness of the proposed search space. For more convincing, **more results** under other splitting settings can be found in Appendix B.

## 5.3 Search Result Analysis

**Architecture Visualization.** The best-found architectures are plotted in Fig. 3. We can find that these two best-found architectures have very similar structures, their encoder parts prefer the convolution operations and FFN modules, especially for their first encoder blocks, while their decoder parts prefer the global operation: the MSHA module, especially for the decoder blocks close to the output, where the convolution operations with large kernel sizes are also preferred. To investigate the observation,

Table 3: The comparison on EdNet dataset in terms of AUC and the number of model parameters: some KT approaches and best architectures found by ENAS-KT with different parameter constraints. SAINT(more) and SAINT+(more) refer to two models with larger hidden and embedding sizes (192).

| Model | Param.(M) | AUC | Model | Param.(M) | AUC | Model | Param.(M) | AUC |
|-------|-----------|-----|-------|-----------|-----|-------|-----------|-----|
| ENAS-KT(2M) | 1.938 | 0.8021 | CT-NCM | 1.997 | 0.7743 | SAINT | 2.749 | 0.7825 |
| ENAS-KT(2.5M) | 2.389 | 0.8047 | SAKT | 2.086 | 0.7650 | SAINT+ | 3.186 | 0.7916 |
| ENAS-KT(3M) | 2.918 | 0.8056 | AKT | 1.233 | 0.7686 | SAINT(more) | **4.910** | 0.7828 |
| ENAS-KT(3.5M) | 3.489 | 0.8059 | NAS-Cell | 1.869 | 0.7796 | SAINT+(more) | **5.627** | 0.7921 |

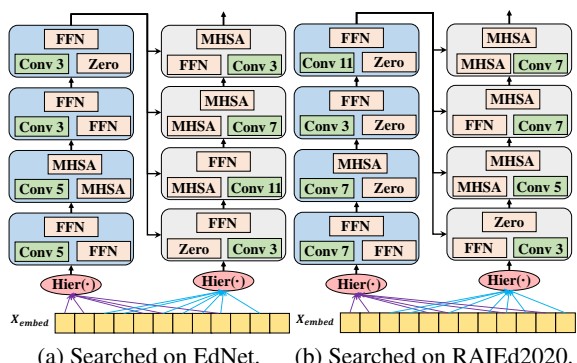

(a) Searched on EdNet.  (b) Searched on RAIEd2020.

Figure 3: The best architectures found on EdNet (a) and RAIEd2020 (b), here we simplify the connections in each encoder/decoder block for better observation.

we further transfer the best architecture found on one dataset to another datasets. The best architecture searched on EdNet dataset achieves an AUC value of **0.8078** on RAIEd2020 dataset, which is very close to the best performance on RAIEd2020 dataset, and the best architecture searched on the EdNet dataset also holds a very promising AUC value on EdNet: **0.8053**. Thus, the best-found architecture holds good transferability and generalization.

**Selected Features.** In addition, their selected input features for encoder and decoder parts are also similar. Fig. 4(a) compares the selected features of SAINT, SAINT+, and the best architecture on EdNet. We can observe that the selected features in SAINT and SAINT+ are a subset of the selected features in the best-found architecture, where the elapsed time embedding and the lag time embedding are considered, and extra features $tag$ and $bund$ are also considered.

### 5.4 Ablation Study

**Effectiveness of Each Component**. To validate the effectiveness of each devised component, four variant architectures (*A-D*) are first given and SAINT+ is taken as the baseline. The comparison results on Ednet dataset have been summarized in Table 4. The variant architecture *A* refers to the vanilla Transformer with all candidate features concatenated and then mapped by a linear transformation as inputs, variant *B* is the same as *A* but only takes the selected features in the best architecture as inputs, variant *C* is the same as *A* but applies the hierarchical fusion method to the selected features, and variant *D* is the Transformer architecture in (11) whose local operations are fixed with $Conv\ 1 \times 3$ and adopts *C*'s input module. The comparison among SAINT+, *A*, *B*, and *C* verifies the effectiveness of the proposed hierarchical fusion method and selected features in the architecture. The comparison between *C* and *D* indicates incorporating convolution operations with Transformer is helpful to improve performance. Moreover, the results of *D* and the best architecture validate the effectiveness of the proposed approach, including the effectiveness of search space as well as the search approach.

Similarly, other four variant architectures (*E-H*) are established in Table 4 to show the influence of each designed component on the best-found architecture's performance. As can be seen, the searched model architecture, the selected features, and the devised fusion module play important roles while the searched model architecture holds the biggest influence on the performance of **Ours**.

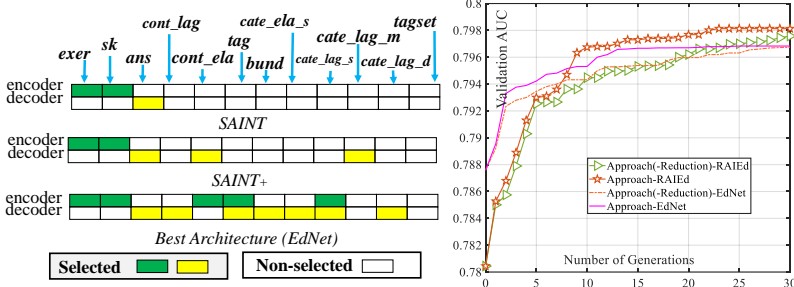

(a) Selected features of the best architecture and SAINT(+).

(b) Convergence profiles of AUC on two datasets.

Figure 4: The comparison of selected features on EdNet (a), and the effectiveness of search space reduction strategy (b).

Table 4: The effectiveness validation of the proposed different components on the EdNet dataset, and the influence of the searched model architecture and the selected features on the best-found architecture's performance is also presented. The Wilcoxon rank sum test with $\alpha$=0.05 is performed.

| Method | RMSE↓ | ACC↑ | AUC↑ | +/-/≈ |
|---|---|---|---|---|
| SAINT+ | 0.4285 | 0.7188 | 0.7916 | 3/0/0 |
| *A*: All Features + Concat | 0.4276 | 0.7203 | 0.7937 | 3/0/0 |
| *B*: Selected Features + Concat | 0.4262 | 0.7217 | 0.7958 | 3/0/0 |
| *C*: Selected Features + Hierarchical | 0.4250 | 0.7236 | 0.7987 | 3/0/0 |
| *D*: *C*'s Input + Convolution | 0.4235 | 0.7253 | 0.8012 | 3/0/0 |
| *E*: **Ours** (without Hierarchical Fusion, with Concat) | 0.4223 | 0.7269 | 0.8041 | 3/0/0 |
| *F*: **Ours** (without the Selected Features, with All Features) | 0.4221 | 0.7260 | 0.8030 | 3/0/0 |
| *G*: **Ours** (without Selected Features & Hierarchical, with SAINT+'s input) | 0.4238 | 0.7249 | 0.8008 | 3/0/0 |
| *H*: **Ours** (without the Searched Architecture, with SAINT+'s model), i.e., *C* | 0.4250 | 0.7236 | 0.7987 | 3/0/0 |
| **Searched** by ENAS-KT(f) (under a small Supernet with fewer training: embedding size 64, epoch 30), retrain under size 128, taking 9.1 hours totally | 0.4224 | 0.7271 | 0.8036 | 3/0/0 |
| **Ours** | **0.4209** | **0.7295** | **0.8062** | - |

To investigate the effectiveness of ENAS-KT with less search cost, as shown in Table 4, a variant ENAS-KT(f) is created. Although its searched architecture's performance is slightly worse than **Ours**, the model performance is better than other approaches and its search cost only takes 9.1 hours, which is much fewer than **Ours** and competitive to NAS-Cell and others (can be found in Appendix B).

**Effectiveness of Search Space Reduction Strategy.** To validate the effectiveness of the devised search space reduction strategy, Fig. 4(b) depicts the convergence profiles of best validation AUC value obtained by the proposed approaches with and without search space reduction strategy on both two datasets, which are denoted by *Approach* and *Approach(-Reduction)*, respectively. As can be seen, the suggested search space reduction strategy can indeed accelerate the convergence of the proposed evolutionary search approach and leads to better final convergence. **More experimental discussions** about the proposed ENAS-KT can be found in Appendix B.

## 6 Conclusion

In this paper, we first introduced the local context modelling held by convolutions to the Transformer, then automated the balancing of the local/global context modelling and the input feature selection by proposing an evolutionary NAS approach. The designed search space not only replaces the original global path in Transformer with the sum of a global path and a local path that could contain different convolutions but also considers the selection of input features. Besides, we utilized an EA to search the best architecture and suggested a search space reduction strategy to accelerate its convergence. Experiments on two largest datasets show the effectiveness of the proposed approach.

## Acknowledgements

This work was supported in part by the National Key R&D Program of China (No.2018AAA0100100), in part by the National Natural Science Foundation of China (No.62302010, No.62107001, No.62006053, No.61876162, No.61906001, No.62136008, No.62276001, No.U21A20512), in part by the Natural Science Foundation of Guangdong Province (Grant No. 2022A1515010443), and in part by the China Postdoctoral Science Foundation (No.2023M740015).

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

## Appendix A   Proposed Approach

**Encoding Strategy**

In the proposed search space, each candidate Transformer architecture can be represented by the following vector-based encoding:

$$\mathcal{A} = [\mathbf{b_{En}}, \mathbf{b_{De}}, (lo \in \{0, \cdots, 4\}, go_1, go_2 \in \{0, 1, 2\})^{1 \times 2N}] =$$

$$\begin{bmatrix} \mathbf{b_{En}} = (b_1, b_2, \cdots, b_i, \cdots, b_{Num}) \in \{0,1\}^{1 \times Num}, 1 \le i \le Num \\ \mathbf{b_{De}} = (b_1, b_2, \cdots, b_i, \cdots, b_{Num}) \in \{0,1\}^{1 \times Num}, 1 \le i \le Num \\ 2N \begin{cases} (lo, go_1, go_2) \\ (lo, go_1, go_2) \\ \cdots \\ (lo, go_1, go_2) \end{cases} \end{bmatrix}, \quad (14)$$

where $Num$ denotes the number of available input embedding and $N$ denotes the number of encoder/decoder blocks. $\mathbf{b_{En}}$ and $\mathbf{b_{De}}$ are used to specify which input embedding are selected for the inputs to encoder and decoder parts, and the triplet $(lo, go_1, go_2)$ is used to specify three operations adopted for each encoder block or each decoder block, i.e., $lo$, $go_1$, and $go_2$ determine which operation is used for $LO(\cdot)$, $GO_1(\cdot)$, and $GO_2(\cdot)$.

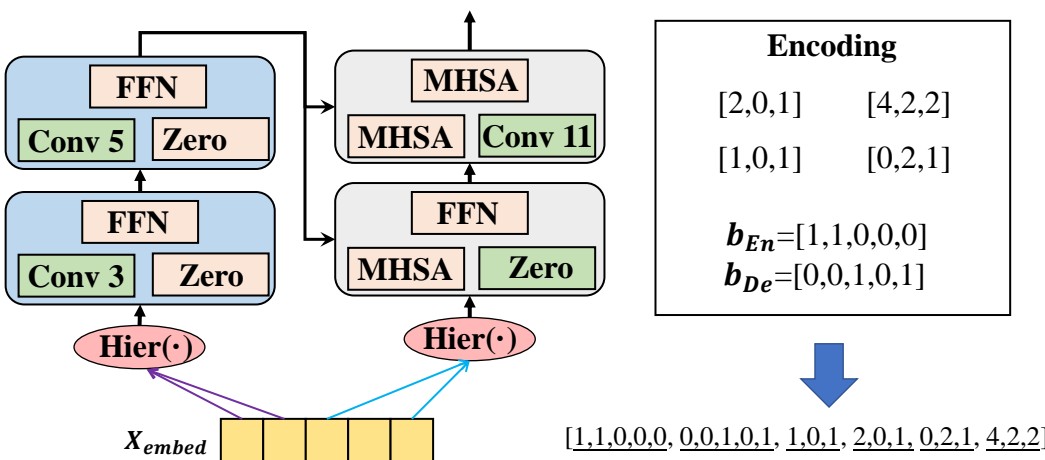

Figure 5: An illustrative example of the utilized encoding strategy.

For a better understanding of the utilized encoding strategy, Fig. 5 gives an illustrative example, where $Num$ and $N$ are set to 5 and 2, respectively. The given architecture takes the first and second embedding as the input to the encoder part, and takes the third and fifth embedding as the input to the decoder part, its two encoder blocks are encoded by $[1, 0, 1]$ and $[2, 0, 1]$, and two decoder blocks are encoded by $[0, 2, 1]$ and $[4, 2, 2]$. As a result, the plotted architecture's encoding is $[1, 1, 0, 0, 0, 0, 0, 1, 0, 1, 1, 0, 1, 2, 0, 1, 0, 2, 1, 4, 2, 2]$.

**Super-Transformer Training**

Inspired by one-shot NAS approaches [20], we use a trained super-Transformer mode to reduce the computational cost, which can provide the weights to directly evaluate the validation AUC values of subsequently sampled sub-Transformer architectures. To make the supernet be trained well, we consider adopting the sandwich training rule [51], which has been the most effective way and widely adopted by one-shot NAS approaches [52, 50]. The original sandwich training rule commonly employs three sub-models for one batch of training, including the largest sub-model, the smallest sub-model, and a randomly sampled sub-model, where three forward passes and one backward pass are executed for one batch update.

**Algorithm 1** Main Steps of ENAS-kT

---

**Input:** $Gen$: Maximum number of generations; $Pop$: Population size; $S$: Search space

**Output:** $\mathbf{P_b}$: The best individual;

1: $\mathbf{P} \leftarrow$ Randomly initialize $Pop$ individuals; % `Population initialization`
2: Evaluate each individual in $\mathbf{P}$ based on the trained super-Transformer and obtain its AUC value;
3: **for** $g = 1$ to $Gen$ **do**
4:    $\mathbf{P}' \leftarrow$ Select $Pop$ parent individuals from Population $\mathbf{P}$; % `Mating pool selection`
5:    $\mathbf{Q} \leftarrow$ Apply the single-point crossover and bit-wise mutation to $\mathbf{P}'$ based on current search space $S$;
6:    Evaluate each individual in $\mathbf{P}$ based on the trained super-Transformer and obtain its AUC value;
7:    $S \leftarrow$ Search Space Reduction$(S, \mathbf{P}, \mathbf{Q})$; % `Reduce search space`
8:    $\mathbf{P} \leftarrow$ Individual Selection$(\mathbf{P} \cup \mathbf{Q})$;
9: **end for**
10: Select the best individual $\mathbf{P_b}$ from $\mathbf{P}$ as the output;
11: **return** $\mathbf{P_b}$ ;

---

To make the training rule adapt the proposed super-Transformer, we utilize three sub-models, including a global sub-model $sub_g$, a local sub-model $sub_l$, and a randomly sampled sub-model $sub_r$, for one batch of training, whose encoding can be represented by

$$\begin{cases} sub_g = [\,\mathbf{b_{En}^r}, \mathbf{b_{De}^r}, (0,2,1), \cdots, (0,2,1), \cdots, (0,2,1)] \\ sub_l = [\,\mathbf{b_{En}^r}, \mathbf{b_{De}^r}, (1,0,0), \cdots, (1,0,0), \cdots, (1,0,0)] \\ sub_r = [\mathbf{b_{En}^r}, \mathbf{b_{De}^r}, (lo, go_1, go_2)^r, \cdots, (lo, go_1, go_2)^r, \cdots, (lo, go_1, go_2)^r] \end{cases} \tag{15}$$

Here $\mathbf{b_{En}^r}$, $\mathbf{b_{De}^r}$, and $(lo, go_1, go_2)^r$ denote these vectors are randomly generated. The global sub-model $sub_g$ denotes the architectures with pure global operations but no local operations, where $\mathbf{b_{en}}$ and $\mathbf{b_{de}}$ are randomly generated, $lo$ is set to 0, $go_1$ and $go_2$ are set to 2 and 1. The local sub-model $sub_l$ refers to the architectures consisting of only convolution operations with the smallest kernel size, where $\mathbf{b_{en}}$ and $\mathbf{b_{de}}$ are also randomly generated, $lo$ is set to 1, both $go_1$ and $go_2$ are set to 0.

As a result, there are three forward passes and one backward pass executed in one batch for updating the super-Transformer. In each batch of training, we employ the Adam [18] optimizer to update the mode parameters via minimizing the binary cross-entropy loss [43] between the predicted response $\hat{r}_i$ and the true response $r_i$:

$$\mathcal{L} = -\sum_i (r_i log \hat{r}_i + (1 - r_i) log(1 - \hat{r}_i)). \tag{16}$$

**Evolutionary Search**

The main steps of the proposed evolutionary search approach are summarized in Algorithm 1, which consists of the following six steps.

First, a population containing $Pop$ individuals will be randomly initialized. During the initialization, it is worth noting that $\mathbf{b_{En}}$ and $\mathbf{b_{De}}$ in each individual have to satisfy the constraints, i.e., $1 \leq sum(\mathbf{b_{En}})$ and $1 \leq sum(\mathbf{b_{De}})$. By doing so, it can ensure there is at least one input embedding for the encoder part and decoder part. Second, the standard binary tournament selection [37] is used to select individual to form the mating pool $\mathbf{P}'$, which randomly selects two individuals from $\mathbf{P}$ each time and keeps the better one in $\mathbf{P}'$ until the size of $\mathbf{P}'$ is equal to $Pop$. Next, select two individuals from $\mathbf{P}'$, and then apply the single-point crossover and the bit-wise mutation [40] to the two individuals to generate two offspring individuals, where the generation process needs to be consistent with current search space $S$. As a result, an offspring population $\mathbf{Q}$ with the size of $Pop$ is obtained. Fourth, the validation AUC values of individuals in offspring population are efficiently evaluated by extracting their needed weights from the trained super-Transformer. Fifth, delete some worse operations from the search space by a devised search space reduction strategy, which makes statistics from the population $P$ and the offspring population $\mathbf{Q}$ and sorts all operations according to the computed fitness. Sixth, all individuals with better validation AUC values in the population $P$ and the offspring population $\mathbf{Q}$ will be kept as new population.

The second to the sixth step will be repeated until the maximal number of generations $Gen$ is exceeded. Finally, the best architecture will be output and then fine-tuned based on the inherited weights from the super-Transformer by minimizing the training loss function in (16).

**Search Space Reduction**

The original search space $S$ has the same length as $\mathcal{A}$, where the item on each bit is an operation set. For example, the item on first bit is $\{0, 1\}$, the items on the bits corresponding to $lo$ are $\{0, 1, 2, 3, 4\}$. To accelerate the convergence of the EA, we further devise a search space reduction strategy to iteratively remove some worse operations from the search space.

Algorithm 2 has summarized the basic procedure. First, population $\mathbf{P}$ and offspring population $\mathbf{Q}$ are combined, two empty lists $SF$ and $SF_{std}$ are used to record each operation fitness and fitness standard deviation of each operation set. Then, get the operation set $Bit_{OpSet}$ on the $i$-th bit of current search space $S$. Third, compute the fitness value of each operation in $Bit_{OpSet}$ by averaging the fitness of the individuals that contain the corresponding operation. Next, $fitness_{OpSet}$, the computed fitness of the set of operations, will be stored as $i$-th item in $SF$, and the fitness standard deviation of $fitness_{OpSet}$ will be also computed and stored in $SF_{std}$. The second to the fourth step will be repeated for all bits in $S$, where $|S|$ denotes the number of bits, i.e., the length. Based on $SF$ and $SF_{std}$, the worst two operations will be deleted, and the worst operation in the operation set with the largest standard deviation will be also removed, where there is at least one operation kept for each bit. By doing so, the ever-reducing search space makes the search process easier.

---

**Algorithm 2** Search Space Reduction

---

**Input:** $S$: Current search space; $\mathbf{P}$: Population; $\mathbf{Q}$: Offspring population;
**Output:** $RS$: Reduced search space;
1: $\mathbf{P} \leftarrow \mathbf{P} \cup \mathbf{Q}, SF \leftarrow \emptyset, SF_{std} \leftarrow \emptyset$;
2: **for** $i = 1$ to $|S|$ **do**
3:    $Bit_{OpSet} = S[i], fitness_{OpSet} \leftarrow \emptyset$; $\%\ OpSet$ : `The operation set on i-th bit`
4:    **for** $op$ in $Bit_{OpSet}$ **do**
5:       $fit \leftarrow$ The fitness value averaged on the individuals containing $op$ in $\mathbf{P}$;
6:       $fitness_{OpSet} \leftarrow fitness_{OpSet} \cup fit$;
7:    **end for**
8:    $SF[i] = fitness_{OpSet}$; $\%$ `Store the fitness of the set of operations on i-th bit`
9:    $SF_{std}[i] = Std(fitness_{OpSet})$; $\%$ `Compute and store the fitness standard deviation for the operation set on i-th bit`
10: **end for**
11: $RS \leftarrow$ Delete the worst two operations from $S$ based on $SF$;
12: $RS \leftarrow$ Delete the worst operation from the operation set with the largest standard deviation in $RS$ based on $SF_{std}$;
13: **return** $RS$;

---

# Appendix B    Experiments

Table 5: Performance comparison under **splitting setting of training and testing of 60% and 30%**. The results of proposed ENAS-KT and compared KT approaches on EdNet and RAIEd 2020 datasets are reported in terms of RMSE, ACC, and AUC, averaged over 5-fold cross-validation. The Wilcoxon rank sum test with $\alpha$=0.05 is also executed.

| Dataset | Metric | DKT | HawkesKT | CT-NCM | SAKT | AKT | SAINT | SAINT+ | NAS-Cell | Ours |
|---|---|---|---|---|---|---|---|---|---|---|
| **EdNet** | **RMSE** ↓ | 0.4661 | 0.4479 | 0.4372 | 0.4411 | 0.4396 | 0.4326 | 0.4288 | 0.4350 | **0.4215** |
| | **ACC** ↑ | 0.6507 | 0.6879 | 0.7048 | 0.6993 | 0.7019 | 0.7128 | 0.7182 | 0.7138 | **0.7289** |
| | **AUC** ↑ | 0.6908 | 0.7475 | 0.7733 | 0.7645 | 0.7683 | 0.7816 | 0.7906 | 0.7782 | **0.8055** |
| **RAIEd2020** | **RMSE** ↓ | 0.4651 | 0.4466 | 0.4357 | 0.4385 | 0.4368 | 0.4313 | 0.4275 | 0.4311 | **0.4207** |
| | **ACC** ↑ | 0.6601 | 0.6905 | 0.7065 | 0.7019 | 0.7076 | 0.7140 | 0.7189 | 0.7164 | **0.7306** |
| | **AUC** ↑ | 0.7057 | 0.7491 | 0.7751 | 0.7665 | 0.7640 | 0.7855 | 0.7927 | 0.7828 | **0.8081** |
| +/-/≈ (six results totally) | | 6/0/0 | 6/0/0 | 6/0/0 | 6/0/0 | 6/0/0 | 6/0/0 | 6/0/0 | 6/0/0 | - |

**Datasets Descriptions**

In our experiments, two largest and most challenging real-world education datasets, EdNet [8] and RAIEd2020 [34], were used. Different common education datasets, such as ASSISTments09 [15],

Table 6: Performance comparison under **splitting setting of training and testing of 50% and 40%**. The results of proposed ENAS-KT and compared KT approaches on EdNet and RAIEd 2020 datasets are reported in terms of RMSE, ACC, and AUC, averaged over 5-fold cross-validation. The Wilcoxon rank sum test with $\alpha$=0.05 is also executed.

| Dataset | Metric | DKT | HawkesKT | CT-NCM | SAKT | AKT | SAINT | SAINT+ | NAS-Cell | Ours |
|---|---|---|---|---|---|---|---|---|---|---|
| **EdNet** | RMSE $\downarrow$ | 0.4664 | 0.4480 | 0.4371 | 0.4423 | 0.4397 | 0.4325 | 0.4286 | 0.4352 | **0.4219** |
| | ACC $\uparrow$ | 0.6503 | 0.6876 | 0.7055 | 0.6986 | 0.7026 | 0.7121 | 0.7182 | 0.7133 | **0.7291** |
| | AUC $\uparrow$ | 0.6902 | 0.7473 | 0.7732 | 0.7640 | 0.7678 | 0.7811 | 0.7903 | 0.7779 | **0.8052** |
| **RAIEd2020** | RMSE $\downarrow$ | 0.4659 | 0.4469 | 0.4354 | 0.4375 | 0.4398 | 0.4315 | 0.4276 | 0.4312 | **0.4200** |
| | ACC $\uparrow$ | 0.6588 | 0.6902 | 0.7068 | 0.7023 | 0.7010 | 0.7146 | 0.7187 | 0.7165 | **0.7309** |
| | AUC $\uparrow$ | 0.7043 | 0.7484 | 0.7759 | 0.7674 | 0.7668 | 0.7851 | 0.7928 | 0.7835 | **0.8085** |
| +/-/$\approx$ (six results totally) | | 6/0/0 | 6/0/0 | 6/0/0 | 6/0/0 | 6/0/0 | 6/0/0 | 6/0/0 | 6/0/0 | - |

JunYi [4], and so on [22, 29], EdNet collected by Santa is currently the largest open available education benchmark dataset, which contains over 90M interactions and nearly 800K students, while RAIEd2020 first appeared in the Kaggle is another publicly available large-scale real-world dataset, which consists of nearly 100M interactions and 400K students.

**More Results under Other Two Splitting Settings**

In previous experiments, all students in each dataset were randomly split into 70%, 10%, and 20% for training, validation, and testing, respectively. Under this splitting setting, the architectures found by the proposed ENAS-KT exhibits better performance than comparison KT approaches.

For more convincing results, we adopted other two settings to split the dataset into training and test datasets for evaluating the model performance. The first one is to split 60% of the dataset as the training dataset and 30% of the dataset as the testing dataset, while the second one takes the splitting ratio of 50% and 40%. The overall comparison results under the two settings are summarized in Table 5 and Table 6, where the average RMSE, ACC, and AUC values over 5-fold cross-validation are reported, the best results are in bold, and the Wilcoxon rank sum test with $\alpha$=0.05 is also executed.

As can be seen from these two tables, the performance of nearly all approaches on both two dataset decrease when the size of the training dataset is reduced while the size of the testing dataset is increased, but some approaches (such as CT-NCM, SAKT, AKT, SAINT+, NAS-Cell, and our approach) hold slightly better performance on RAIEd2020 under the splitting setting of 50% and 40% than that of the splitting setting of 60% and 30%. Under whichever splitting setting, the architectures found by the proposed approach still outperform the state-of-the-art KT approaches, which can be seen from the statistical results of the Wilcoxon rank sum test.

**Selected Features**

Due to page limit, previous experiments do not show the selected features of the best-found architecture on the RAIEd2020 dataset. For a more comprehensive observation and comparison, Fig. 6 presents the selected input features of four models, i.e., SAINT, SAINT+, best-found architecture on EdNet, and best-found architecture on RAIEd2020. It can be found that the selected features in SAINT and SAINT+ are a subset of the selected features in both two best-found architectures, where the elapsed time embedding and the lag time embedding are considered in two est-found architectures, and extra features (such as $tag$, $bund$, and $tagset$) are also considered. Besides, we can also find that the selected input features of these two found architectures are very close, where there is only one input feature difference (i.e., $cate\_lag\_d$) between their encoder parts, and only two input features ($bund/expla$ and $tagset$) are different between their decoder parts. This implies that the selected input features are more important for our Transformer architectures to solve the KT task than those not selected by the two architectures.

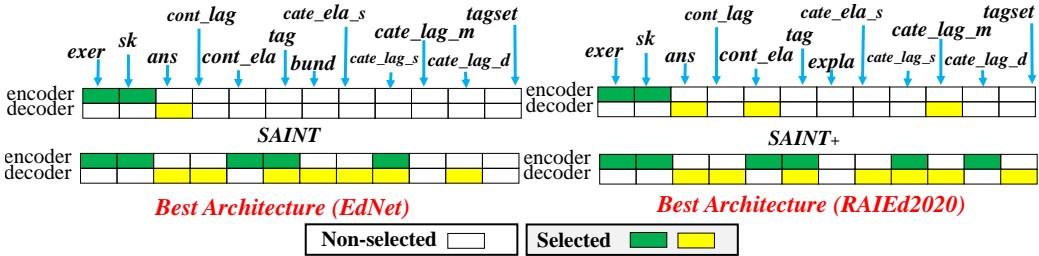

Figure 6: The visualization of selected input features of SAINT, SAINT+, and two best-found architectures on EdNet and RAIEd2020.

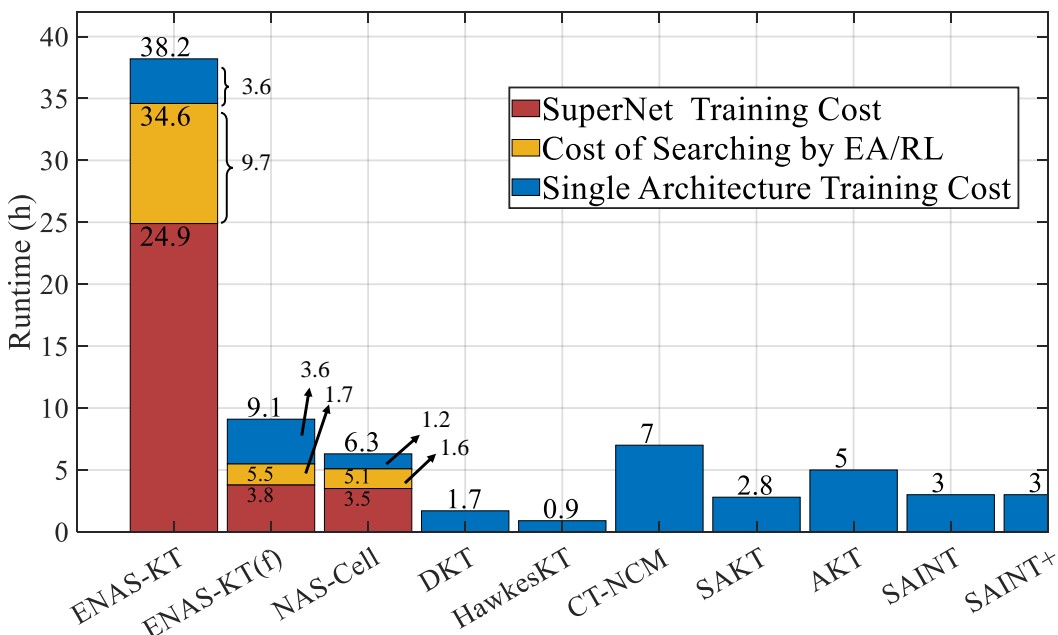

Figure 7: The runtime comparison between ENAS-KT, ENAS-KT(f) and KT approaches on Ed-Net, which reports the cost of searching, supernet and single architecture training for NAS-based approaches.

## Limitation Discussion

The search cost of neural architecture search (NAS)-based approaches are always the focus of researchers, especially when they are compared to non-NAS-based approaches. To investigate this, Fig. 7 compares the runtimes of the proposed ENAS-KT, its variant approach ENAS-KT(f) and comparison KT approaches. Here the runtimes of ENAS-KT, ENAS-KT(f) and NAS-Cell contain the cost of supernet training, architecture searching and single architecture retraining, while runtimes of other KT approaches are only used for training their models.

It can be observed that the training cost of the architecture found by the proposed ENAS-KT is 3.6h, which is better than CT-NCM and AKT, worse than NAS-Cell, DKT, and HawkesKT, and competitive to other KT approaches. Besides, the search cost of the proposed ENAS-KT is 34.6h, leading to a total cost of 38.6h for ENAS-KT, which is higher than that of NAS-Cell (6.3h). This is because Transformer architectures in the proposed approach are more complex than that of NAS-Cell and thus need more time to train them. However, considering the significant performance leading of the proposed ENAS-KT over comparison approaches, the high overall cost of the proposed approach is acceptable for the KT task. Moreover, note that ENAS-KT(f) is a variant of the proposed ENAS-KT,

which searches for architectures under a smaller supernet with fewer training epochs. Specifically, the embedding size is halved to 64 for the supernet, the number of training epochs is also halved to 30, but the best-found architecture holds the same settings as **Ours** for a fair comparison. As shown in Table 4, the architecture found by ENAS-KT(f) holds a 0.8036 AUC value. Although its performance is worse than ours, as shown in Fig 7, its overall cost only takes 9.1 hours (3.8hours for supernet training, 1.7 hours for evolutionary search, and 3.6 hours for final training), which is fewer than the proposed ENAS-KT and competitive to NAS-Cell and others.

In summary, the current settings adopted in the proposed approach provide significantly better performance with a much higher cost, but our approach can also achieve competitive results at a much lower cost, whose performance is still better than other KT approaches and whose cost is competitive. In future work, we would like to explore surrogate models in the proposed approach to further reduce the search cost.

In addition to the search cost, the validity of the proposed approach also suffers from one external threat, i.e., too many available input features in new datasets ($Num$ is large). The hierarchical feature fusion manner in Equation 10 utilizes pairwise concatenation, which requires a large number of parameters. When the datasets hold too many input features, the number of feature pairs will increase exponentially, e.g., 10 features need 45 concatenation operations while 20 features need 190 concatenation operations, which further causes a high increase in the number of network parameters and deteriorates the models' efficiency. Therefore, the validity and efficiency of the proposed approach will be highly affected by the number of available input features in new datasets. But because nearly all known education datasets hold fewer available features, the proposed approach is effective and can counter most datasets in KT. In future work, we would like to design novel fusion modules to aggregate features without considering the influence of datasets to be handled.

