# OpenReview forum: "Evolutionary Neural Architecture Search for Transformer in Knowledge Tracing"
_NeurIPS.cc/2023/Conference — NeurIPS 2023 poster_

### Official Review · Reviewer_EGZn · 2023-07-02

**Soundness:** 3 good
**Presentation:** 3 good
**Contribution:** 3 good
**Rating:** 7
**Confidence:** 4

**Summary:**

This paper proposes to search for the optimal Transformer-based architecture for the knowledge tracing task. The authors have effectively constructed a search space that captures both global and local context modeling to accurately capture students' forgetting behavior. To further improve the efficiency of the model, a selective hierarchical input module has been designed to automate feature selection, eliminating the need for manual selection. By training the supernet and adopting the EA search, the authors have achieved superb results in two education datasets, EdNet and RAIEd2020.

**Strengths:**

- The paper presents a clear motivation for the study, emphasizing the novel approach of utilizing NAS techniques to solve problems in KT, which is worth exploring and bridges the gap between KT and NAS
- The paper is well-structured and easy to follow, making it accessible to readers.
- The design of the search space for KT is reasonable and effective. And the experimental results seem convincing.

**Weaknesses:**

- The search cost should be reported, including the supernet training time and the evolutionary search time.
- The parameters and FLOPs of all architectures in Table 2 should be reported to provide a comprehensive understanding of the effectiveness.
- As an expert in NAS, I appreciate the authors' efforts in customizing an effective search space for KT and conducting the search for better architectures, which is a reasonable approach. However, from my perspective, it's difficult to judge the improvement shown in Table 2 without considering the change in parameters and FLOPs, which are necessary for a comprehensive evaluation.

**Questions:**

None

---

> ### Author Rebuttal · Authors · 2023-08-08
>
> ## Response to Weakness 1
> Although we have reported the search cost of our approach and compared approaches in  **Figure 3** of ***the previous Appendix***,
> Figure 3 only reports each approach's overall cost and the training cost of the final architecture,
> where the cost of the supernet training and the evolutionary search are not presented for *ENAS-KT*.
>
> **According to the reviewer's suggestion**, we have replotted Figure 3 as **Fig.1(a)** in  ***global_response***, which reports the supernet training time,  the evolutionary search time,  and the final architecture training time for the proposed approach *ENAS-KT*.
>
> - As can be seen from  **Fig.1(a)**, for the proposed *ENAS-KT*, its final architecture training takes 3.6 hours, the evolutionary search takes 9.7 hours, while the supernet training takes 24.9 hours, which accounts for a large proportion of its overall cost.  The computational cost of the proposed approach is higher than that of the traditional KT approaches (exclude *NAS-Cell*) since they are not search-based approaches, and their cost is only used to train one architecture in these approaches.
>
> - Even compared to *NAS-Cell*, the cost of the proposed approach is also non-competitive since the supernet in our approach is more complex and needs more epochs for training. To validate the effectiveness of  *ENAS-KT* under a smaller supernet trained for fewer epochs,  a variant **ENAS-KT(f)** is created. In **ENAS-KT(f)**, the embedding size $D$ is halved to 64 for the supernet, the number of training epochs is also halved to 30, but its best-found architecture holds the same settings as **ours** for a fair comparison.
>
> - The comparison results have been summarized in **Table II**  of   ***global_response***, the best architecture found by **ENAS-KT(f)** is denoted as **Searched** and holds a 0.8036 AUC value.
> Although its performance is worse than **ours**, as shown in **Fig.1(a)** of   ***global_response***, its overall search cost only takes **9.1 hours** (**3.8hours for supernet training**, **1.7 hours for evolutionary search**, and 3.6 hours for final architecture training),
> whose **search cost (i.e., 3.8+1.7 = 5.5 hours)** is significantly smaller than **ours** (34.6 hours) and competitive to *NAS-Cell* (5.1 hours).
>
> In summary, the proposed approach under the original search settings indeed holds an extremely high computational cost due to its high cost for supernet training and evolutionary search, but the proposed approach can achieve a competitive computational cost with a bit expense of the final architecture performance.
>
>
>
> ## Response to Weakness 2
> We agree with the reviewer that comparing the parameters/FLOPs of networks is a very common metric for experiments in NAS,
> which is meaningful but commonly ignored in KT approaches' experiments.
>
> As suggested by the reviewer, we would like to compare the parameters and FLOPs of all architectures in the original Table 2,
> but we cannot find a suitable Python package to compute the  FLOPs for KT models because KT models' inputs are not as simple as CNNs' inputs that have a fixed form, which causes existing packages for FLOPs (such as fvcore, thop, and torchstat)  inapplicable.
> Besides, it is too complex for us to calculate FLOPs for each model manually.
>
> - Therefore, we only compared the number of model parameters and added the comparison to the original Table 2, forming **Table I** in ***global_response***. Since the number of parameters for KT models is related to inputs,  **Table I** presents the model parameter information for each KT approach on EdNet and RAIEd2020.
>
> - As can be seen, *HawkesKT* holds the fewest model parameters, *DKT* is the second-best, and the architecture found by the proposed *ENAS-KT* holds the most model parameters, which is worse than other KT models. Actually,  the high model parameters of **Ours**  are attributed to its input part, which is used to get the input embeddings and aggregate these embeddings.
>
> - For an intuitive observation, we further plotted  **Fig.1 (b)** in ***global_response*** to depict the number of parameters comparison between ENAS-KT and some KT approaches in terms of the **model main body** and **input part**. Here *the input part* refers to the parts in  KT models to get the embeddings from inputs and aggregate these embeddings, while *the model main body*  refers to other parts in KT models excluding *the input part*.
>
> - As can be observed, *the input part* of ENAS-KT  is the most complex, holding about 2.5 M parameters, which is worse than other KT models and only competitive to that of SAINT+.
> It is reasonable because the architecture of  ENAS-KT has more input features to handle and contains a complex **hierarchical fusion module**. However,  in terms of *the model main body*, ENAS-KT   is competitive to SAINT, SAINT+, and AKT.
> Especially for AKT, the number of its overall model parameters is about 1.23M, much better than ENAS-KT,
> but the parameters in its  *the model main body* are similar to  ENAS-KT.
>
> In summary,  the architecture found by our approach indeed holds more parameters, which is mainly caused by the input part.
>
> ## Response to Weakness 3
> Thanks for your recognition of our work.  According to your suggestions, we have added the model parameter comparison to the original Table 2 (forming **Table I** in ***global_response***) for comprehensive comparison, and we also presented a figure (**Fig.1 (b)** in ***global_response***) for a deep insight into the model parameters.
>
> To sum up,  the proposed approach is non-competitive to other KT approaches in terms of the model parameters, but our *the model main body* is competitive to that of Transformer-based KT approaches because high parameters in our approach are attributed to *the input part*. As a result, it is acceptable for us to increase the model performance with a bit increasing of model parameters.
>
> ### We appreciate your valuable feedback and will revise the paper based on the above.

---

> > ### Comment · Reviewer_EGZn · 2023-08-12
> > **Discussion about the effectiveness.**
> >
> > Thanks for the authors' efforts and feedback. The detailed search cost has addressed my first question. However, I still have some doubts about the effectiveness of this method. As shown in Table I of the global_response, this method uses significantly more parameters than related works, making it difficult to distinguish whether the performance improvement is due to the search space design or the increase in model parameters. Since the EA method is used to search for the final architecture, I suggest that the authors add parameter constraints to search for several architectures with different model parameters, which can create a Pareto-front to compare with others in a fairer way.

---

> > > ### Author Response · Authors · 2023-08-14
> > > **Further validation of the effectivness.**
> > >
> > > According to the reviewer's suggestion, we have executed the EA multiple times on the trained super-Transformer to search for several architectures with different model parameters.
> > >
> > >  Due to the parameter limitation of the input feature selection part to the models (the input part holds at least 1.7 M parameters),
> > > four parameter constraints (including 2M, 2.5M, 3M, and 3.5M) were used in the EA for searching on the EdNet dataset,  respectively, where the parameters of all architectures generated in the EA cannot exceed the used parameter constraint.
> > >
> > > As a result, four architectures were found by the EA, which are denoted as **ENAS-KT(2M)**, **ENAS-KT(2.5M)**, **ENAS-KT(3M)**, and **ENAS-KT(3.5M)**.  The following is a brief comparison between architectures found by ENAS-KT and some  KT models  in terms of AUC values and the number of parameters on the EdNet dataset:
> > > | **Model** | **Parameters(M)**| **AUC** | **Model** | **Parameters(M)**| **AUC**|
> > > |---|:---:|:---:| ---|:---:|:---:|
> > > |**CT-NCM**|  1.9974| 0.7743| **ENAS-KT(2M)**| 1.9377 | 0.8021 |
> > > |**SAKT**|2.0864  |0.7650 | **ENAS-KT(2.5M)**| 2.3885 | 0.8047|
> > > |**AKT**| 1.2330 |0.7686 | **ENAS-KT(3M)**| 2.9179 | 0.8056|
> > > |**SAINT**| 2.7492  | 0.7825 | **ENAS-KT(3.5M)**| 3.4896 | 0.8059|
> > > |**SAINT+**| 3.1862 |0.7916 | **ENAS-KT**|  3.8232| 0.8062|
> > > |**NAS-Cell**| 1.8692 |0.7796 | |  | |
> > >
> > > As can be seen, the proposed approach ENAS-KT can still find good potential architecture under the parameter constraint.
> > > - Specifically, when under the constraint of 2M, most candidate operations are available, but the available input features are greatly restricted, only the exercise and response could be selected.  As can be seen from **Table II** in ***global_response***, the model main body architecture plays a more important role than the input part, and thus only searching for the model main body architecture is also effective. That can explain the good potential performance of  **ENAS-KT(2M)**.
> > > - When the parameter constraint is relaxed to 2.5M and 3M, most candidate operations and most candidate input features become available,  thus it can be seen there is a big performance improvement of **ENAS-KT(2.5M)** and **ENAS-KT(3M)** compared to  **ENAS-KT(2M)**.
> > > - When the parameter constraint is further relaxed, as can be seen from  **ENAS-KT(3.5M)**  and **ENAS-KT**, the performance improvement becomes less and less.
> > >
> > > We can indeed see that the found architecture's performance increases with model parameters increasing,
> > > but the increase in model parameters is caused by the change of the found model architecture, not the embedding or hidden size.
> > > Therefore, we can draw a basic conclusion that the effectiveness of the proposed approach is mainly not attributed to its high model parameters,  to some extent, which can be mainly attributed to the suggested search space.
> > >
> > > To further validate this, we also executed an indirect experiment that we have ever tried:  trying to improve the performance of **SAINT** and **SAINT+** by increasing their model parameters.
> > > Here, we increase their embedding and hidden sizes to 192 (128 original), leading to **SAINT(more)** and **SAINT+(more)**.
> > > The following is a basic comparison with **ENAS-KT**:
> > > | **Model** | **Parameters(M)**| **AUC** |
> > > |---|:---:|:---:|
> > > |**SAINT**| 2.7492  | 0.7825 |
> > > |**SAINT(more)**| **4.9102**| 0.7828 |
> > > |**SAINT+**| 3.1862 |0.7916 |
> > > |**SAINT+(more)**| **5.6271**|0.7921 |
> > > | **ENAS-KT**|  3.8232| 0.8062|
> > >
> > > As can be seen, **SAINT(more)** and **SAINT+(more)** hold nearly two times of model parameters as **SAINT** and **SAINT+**, respectively, but their performance improves very little, with nearly no improvement. As a result,  increasing the model parameters for **SAINT and SAINT+  to improve their performance does not work and is unworthy.
> > >
> > > In such a context, we can draw the conclusion that only increasing the model parameters in the proposed approach may not have much effect on its performance improvement.
> > >
> > > ### To sum up, we think the high performance of the final architecture found by our approach is mainly attributed to the search space design, not mainly affected by the increase in model parameters.
> > >
> > > ### We appreciate your valuable feedback again, and thank you very much.

---

> > > > ### Comment · Reviewer_EGZn · 2023-08-14
> > > > **Thanks for conducting those additional experiments.**
> > > >
> > > > The provided results are robust and sufficient to demonstrate the effectiveness of this method. Please include these experiments in the final version of the paper. I would like to raise my score and suggest authors to release their code upon acceptance.

---

> > > > > ### Author Response · Authors · 2023-08-14
> > > > > **Many thanks for your recognition.**
> > > > >
> > > > > We will add the experiments to these final version, and it is our pleasure to release our code.
> > > > >
> > > > > Thank you to take the time to give insightful suggestions.

---

### Official Review · Reviewer_XrVw · 2023-07-03

**Soundness:** 3 good
**Presentation:** 3 good
**Contribution:** 2 fair
**Rating:** 5
**Confidence:** 2

**Summary:**

This paper modifies the Transformer architecture and employs Neural Architecture Search (NAS) to tackle knowledge tracing tasks

**Strengths:**

Writing is good and the method looks sophisticated.

**Weaknesses:**

As I am not specialized in the field of knowledge tracing, I cannot accurately assess the novelty of this work or the adequacy of the experiments. However, as a general ML/NLP researcher, I can offer some intuition:

1. If the architecture in the Transformer is modified, then the pre-trained model is unavailable. Would it be necessary to pre-train the model from scratch.

2. NAS has been extensively researched. Although this paper appears to be the first to apply NAS specifically to the knowledge tracing problem, it is not clear why NAS is considered the most suitable technique for this task. For instance, have the authors considered alternative approaches such as mask-based methods [1]? Providing a comparison or justification would strengthen the argument for utilizing NAS in this context.

[1]: Continual Training of Language Models for Few-Shot Learning. EMNLP 2022


**Questions:**

See above

**Limitations:**

I don't see the discussion of limitations.

---

> ### Author Rebuttal · Authors · 2023-08-09
>
> ## Response to Weakness
>
> We would like to clarify the novelty of our work.  While it is true that NAS has been extensively researched, its application in the field of KT is indeed novel. The main challenge in KT is to model the students' knowledge states accurately, which involves capturing both the global and local context of students' learning behavior. Traditional Transformer-based KT approaches have limitations in capturing the local context, especially the forgetting behavior of students when the related records are distant in time. Our work addresses this issue by introducing convolution operations to the Transformer, enhancing its local context modeling ability. Furthermore, we propose an evolutionary NAS approach to automate the input feature selection and determine where to apply which operation for balancing the local/global context modeling. This is a good improvement in the field of KT.
>
> ## Response to Weakness 1
>
> My answer is yes. Due to the difference between model architecture, the model in our approach can not make use of existing pre-trained language models to initialize the model's weights for a warm start.
>
> - You are correct, it is the best choice to train the supernet model from scratch when the proposed approach is used for new KT datasets.
>  The reason behind this is that the difference in KT datasets' input features causes the difference in the supernet's input part, and thus a supernet (whose weights) for a specific KT dataset cannot be directly used as the supernet for other KT datasets.
>
> - However, in addition to training the supernet model from scratch,  once there has been a supernet trained for one KT dataset,  it could train the supernet model from the existing trained supernet for another KT dataset.  The supernet for another KT dataset could inherit the weights of the model part of the trained supernet (excluding the input part's weights) for a good initialization, and then fine-tune the supernet to get a well-trained model.
>
> - Besides, for  all sub-models in the same KT dataset, these sampled  sub-models do not need training from scratch, because they could inherit their needed weights from the trained supernet for continued training,
> which is equal to the fine-tuning process.
>
> - Furthermore, the main purpose of our work is to enhance the Transformer's ability to model local context, which is crucial for capturing students' forgetting behavior in knowledge tracing. This improvement is aimed at addressing challenges specific to knowledge tracing problems, which may not have been adequately considered in pre-trained models.
> Therefore, despite the need to start training from scratch sometimes, we believe it is worth it.
>
>
> ## Response to Weakness 2
>
> Regarding your question about the suitability of NAS for this task, we believe NAS is a powerful tool for our problem for several reasons.
> - As validated in previous KT research, e.g., SAINT, SAINT+,  the input features of the KT model play important roles in enhancing KT performance and improving the accuracy of tracing student's proficiency.  However, manually selecting suitable features as the KT model's inputs is labor-intensive,  and general feature selection strategies（such as embedded-based feature selection strategies）are also inapplicable, needing substantial computational costs. Thus it is necessary to propose an effective yet efficient method to find suitable features for KT models. NAS is designed to automate the process of architecture selection, which aligns with our goal of automating input feature selection in KT.  Therefore, taking the feature selection task as a NAS task and solving it with a super-based NAS approach is feasible and effective.
>
> - Traditional Transformer-based KT approaches have limitations in capturing the local context. We aim to introduce convolution operations to the Transformer, enhancing its local context modeling ability. However, finding an architecture that can balance the global and local context modeling abilities in the KT model well is difficult. In such a context,  NAS allows us to explore a vast search space of possible architectures, which is crucial for finding an optimal solution that can balance local and global context modeling in KT.
>
> We apologize that we are unfamiliar with mask-based methods. After we read the reference (CPT) you suggested, we think such a method is similar to but different from our approach. The following is our understanding (maybe not quite right):
> - The CPT uses a "mask-based method". Its basic idea is to introduce a special mask in the input of the model, which can be used to indicate which parts of the input of the model are related to the current task and which parts are related to other tasks.  In this way, the model can only pay attention to the input part related to the new task while ignoring the input part related to the old task when processing the new task. In this way, the model can effectively learn new tasks without forgetting old tasks
>
> - The above process of masking some inputs of the model for a new task can be seen as sampling a sub-model from the supernet, where only the inputs and architectures related to the sampled sub-model are active while others are masked. The fine-tuning of CPT for a new task is basically equal to the fine-tuning of the sub-model in our approach.
>
> - However,  in CPT, it is known which parts for a new task need to be masked, but in our case, it is unknown which input features and which architectures are useful, which needs extra searching to find the best combination of input features and model architectures.
>
>
> Nevertheless, we agree that a comparison or justification for the choice of NAS would strengthen our argument. In the revised version of our paper, we will provide a more detailed discussion on why we choose NAS over other methods, including mask-based methods.
>
> ### We appreciate your valuable feedback and will make the corresponding revision based on the above.

---

### Official Review · Reviewer_YAb1 · 2023-07-05

**Soundness:** 3 good
**Presentation:** 4 excellent
**Contribution:** 3 good
**Rating:** 6
**Confidence:** 5

**Summary:**

This paper presents neural architecture search (NAS) for transformer in the context of educational application, specifically knowledge tracing. The authors employed two level NAS:
one at a local level to encorporate forgetting behavior and architecture serach at global level for exercise and response related embedding. The authors evaluated proposed approach
on two large datasets and compared with eight SOTA KT approaches.

**Strengths:**

1. Local path for forgetting behavior encoding in KT
2. NAS for KT.

**Weaknesses:**

Comparison with other neural architecture search methods is not discussed. Please see Reference [1] for NAS experiment settings.
The technical contribution of the paper seems rather limited.
How does the proposed NAS generalize to other transformer-based KTs, such as SAKT and AKT?

References.
1. Ding M, Lian X, Yang L, Wang P, Jin X, Lu Z, Luo P. Hr-nas: Searching efficient high-resolution neural architectures with lightweight transformers. In Proceedings of the IEEE/CVF conference on computer vision and pattern recognition 2021 (pp. 2982-2992).

**Questions:**

How does the proposed NAS generalize to other transformer-based KTs, such as SAKT and AKT?

**Limitations:**

Yes, the trade-off between the time complexity of NAS is discussed in the Supplementary.

---

> ### Author Rebuttal · Authors · 2023-08-09
>
> ## Response to Weakness 1
> Actually, we have discussed the comparison of the proposed approach ENAS-KT and existing NAS approaches for other tasks in terms of approach design in the *Related work* of the submitted paper, where the approach design comparison between ENAS-KT and NAS-Cell is also presented.
>
> But we have to admit that there lacks a discussion about the comparison between the proposed ENAS-KT and some NAS approaches for KT tasks, especially in terms of model performance, model complexity, and overall search cost.
>
> We would like to give a comprehensive and detailed comparison like the reference [1] you suggested by comparing our approach with many NAS approaches.
> However, due to the rarity of NAS-based KT approaches (only one approach NAS-Cell), the following will discuss the comparison between the proposed ENAS-KT and NAS-Cell from three perspectives:
> 1. - **In terms of approach design**.   Up to now,  NAS-Cell is the first and only method to apply NAS to KT.  But NAS-Cell is simple and lacks the design for KT tasks. It directly uses ENAS[R1]  to search the optimal LSTM cell for  KT.  As you know, ENAS is a one-shot NAS based on reinforcement learning, which trains a supernet for all subsequently sampled sub-models to obtain their performance directly.
> - It can be seen that there is no KT-specific design in NAS-Cell, which still searches for the general LSTMs.
> Instead, our approach not only devises the input feature selection part to select suitable input features but also modifies the Transformer's architecture to achieve the balance between the global and local context modeling abilities.
> In short, our approach is well-designed for KT tasks, which can improve the model performance highly.
>
> 2. - **In terms of model performance and complexity**.  To compare this, we have summarized all approaches' performance and model parameter information in **Table I**  of ***global_response***. （Since the number of parameters for KT models is related to inputs,  **Table I** presents model parameter information for each KT approach on EdNet and RAIEd2020.）
>  Besides, for a deep insight into the model parameter, we further plotted  **Fig.1 (b)** in ***global_response*** to depict the model parameter comparison between ENAS-KT and KT approaches in terms of the **model main body** and **input part**. (Here *the input part* refers to the parts in  KT models to get the embeddings from inputs and aggregate these embeddings, while *the model main body*  refers to the other parts in KT models excluding *the input part*.)
>
> - As can be seen, the architecture found by the proposed *ENAS-KT* holds the most model parameters, which is worse than other KT models, especially worse than NAS-Cell.  Besides, as can be seen from **Fig.1 (b)**, although the high model parameters of **Ours**  are attributed to its input part,  the model parameters in its *the model main body*  are still worse than NAS-Cell, only competitive to SAINT, SAINT+, and AKT. Therefore,  the architecture found by our approach indeed holds significantly better performance while holding more model parameters compared to NAS-Cell.
>
> 3. - **In terms of the search cost**.  To this end, **Fig.1** in ***global_response*** reports the cost for each KT approach, where the supernet training time, the search time, and the final architecture training time for *ENAS-KT*, *ENAS-KT(f)*, and  NAS-Cell are also elaborated.  Here *ENAS-KT(f)* is a smaller variant of *ENAS-KT* to show that our approach is able to final a sufficiently good architecture at a much lower cost. (*ENAS-KT(f)* uses a smaller supernet with less training:  size $D$ is halved to 64, and training epochs are also halved to 30, but its best-found architecture holds the same settings as *ENAS-KT* for a fair comparison).
>
> - As can be seen, *ENAS-KT* takes 3.6 hours, 9.7 hours, and 24.9 hours for final training,  evolutionary search, and supernet training, respectively. Its overall search cost is higher than all KT approaches including *NAS-Cell*. This is because the supernet in *ENAS-KT* is more complex and needs more epochs for training.  However,  as shown in **Table II** and **Fig.1(a)**, when *ENAS-KT* is with a smaller supernet (i.e., *ENAS-KT(f)*),  its performance is still sufficiently good and better than *NAS-Cell* but its search cost is competitive to *NAS-Cell*.
>
>  In summary, the proposed approach is better than *NAS-Cell* regarding approach design and model performance, competitive to    *NAS-Cell* regarding search cost, but worse than  *NAS-Cell* regarding mode complexity.
>
> [R1] Pham H, et al. Efficient neural architecture search via parameters sharing[C]//International conference on machine learning. PMLR, 2018.
>
> ## Response to Weakness 2 (Question 1)
> The proposed approach cannot be directly applied to SAKT and  AKT,  because our search space (including the input feature selection part and the model main architecture part) is inapplicable to them, especially for the model main architecture part, which is not original Transformer.
>
> However, from the comparison between SAINT+ and $\mathbf{C}$ in **Table II**, we think that the input feature selection part could be used for SAKT and  AKT to select their suitable input features to improve performance.
>
> To this end, we executed the same supernet training and evolutionary search procedure for SAKT and AKT, and obtained their own best architectures, denoted by SAKT(NAS)  and AKT(NAS). The following is the comparison results:
> |AUC |SAKT |SAKT(NAS) |AKT |AKT(NAS)|
> |----|----|----|----|----|
> |EdNet| 0.7650|0.7898| 0.7686|0.7791|
> |RAIED2020| 0.7693|0.7925 |0.7752|0.7873|
>
> As can be seen, the performance of both SAKT and AKT is improved but the improvement of AKT is lower than that of SAKT. This may be because the input form of SAKT is close to ours, but the query computation manner for AKT from its input is unique and different from ours.
>
> ### We appreciate your valuable feedback and will revise the paper based on the above.

---

> > ### Comment · Reviewer_YAb1 · 2023-08-20
> >
> > I would like to thank the authors for their response and additional experiments. I have updated my review score to Weak Accept.

---

> > > ### Author Response · Authors · 2023-08-21
> > > **Many thanks for your recognition.**
> > >
> > > We greatly appreciate you taking the time to give insightful questions and read the response. Thanks a lot!

---

### Official Review · Reviewer_YzcD · 2023-07-06

**Soundness:** 3 good
**Presentation:** 2 fair
**Contribution:** 3 good
**Rating:** 5
**Confidence:** 3

**Summary:**

This paper introduces neural architecture search (NAS) for Transformer in the field of Knowledge Tracing (KT) for the first time. The authors propose a Transformer architecture that combines local and global paths, as well as a search space, to address the issue of students’ forgetting behavior in the knowledge tracing field. In this process, they propose a hierarchical fusion method that selects and fuses embeddings from various features. The proposed model demonstrates outstanding performance in experiments conducted on two large knowledge tracing datasets.


**Strengths:**

- The attempt to apply Transformer NAS in the field of knowledge tracing is innovative. Moreover, the search space they introduced to apply Transformer NAS in the KT field is convincingly related to students' forgetting behavior.

- To reduce the high computational cost of evolutionary algorithms, the method used by the authors efficiently decreased the computational cost to a competitive level.


**Weaknesses:**

- Hierarchical fusion is excessively large architectures. The authors suffered from high computational costs and had to limit the number of feature candidates (Appendix 2, Limitation Discussion). While the proposed fusion method is intriguing, it would be beneficial to conduct more experiments on datasets with a large number of features, datasets with limited training data, or specifically focusing on the hierarchical fusion method.

**Questions:**

- The reported ablation study (Table 3) does not demonstrate the dependencies between each component. It would be beneficial to describe the contribution of each component to the final model's performance. In particular, it would be valuable to know whether the performance improvement is primarily due to the model architecture (Figure 3) or the feature selection (Figure 4a).

- Figure 2 appears to require clarification. In particular, the meaning of the figure regarding “search space reduction” is unclear.

- The proposed model requires a computational cost that is at least 5 times higher compared to existing models, with most of the cost attributed to the challenging super-Transformer training cost, which is difficult for users to optimize (Appendix, Figure 3). Is this cost an inherent limitation of proposed evolutionary NAS? Can sufficiently competitive results be achieved even with lower costs (e.g., smaller architecture, fewer training steps)?


**Limitations:**

- The authors well summarized the limitations of the paper.

---

> ### Author Rebuttal · Authors · 2023-08-08
>
> ## Response to Question 1
> According to your suggestion, we have added more variants to the original Table 3 (forming **Table II** of   ***global_response***) to show the effectiveness of searching and the contribution of each component to the final model's performance.
>
> Firstly, two variant architectures $\mathbf{H}$ and $\mathbf{I}$ are created to answer the reviewer's main concerns:
> -  $\mathbf{H}$ is our final model without the input feature selection part, which directly takes  all available features as input features and aggregates the features as the  model's input  by **the simple sum operation**;
> - $\mathbf{I}$ is our final model without the model architecture found in Figure 3, which employs the vanilla Transformer architecture (same as SAINT+) as the model architecture. Here $\mathbf{I}$ is equal to $\mathbf{C}$ in the submitted paper.
>
> As can be observed from the results of $\mathbf{H}$ and $\mathbf{I}$ in  **Table II** of   ***global_response***, both the model architecture part and the hierarchical feature selection part play important roles in performance improvement, but the model architecture part is more important, whose performance improvement on AUC  is more than that of the hierarchical feature selection part by 0.002.
>
> Secondly, we  added  two more  variant architectures  $\mathbf{F}$ and $\mathbf{G}$ to analyze the contribution  of the selected features and the hierarchical fusion module  to the final model's performance:
> - $\mathbf{F}$  denotes our final model without the hierarchical fusion module, which directly utilizes the concatenation operation to aggregate the embeddings of selected features;
> - $\mathbf{G}$  denotes our final model without using the selected features, which directly takes all available features as input features and aggregates them by the hierarchical fusion module.
>
> As can be seen from the results of $\mathbf{F}$ and $\mathbf{G}$, both the selected features and the hierarchical fusion module are effective in improving the final model's performance,  and the selected features are more important than the hierarchical fusion module.
> **PS**: the importance of the two parts changes when applied to vanilla Transformer from the results of $\mathbf{A}$, $\mathbf{B}$, and $\mathbf{C}$.
>
> Besides, we also added two architectures' results to **Table II** of  ***global_response*** to show the effectiveness of evolutionary search:
> -  "Super-Transformer (after training)" refers to the performance of the super-Transformer after training. The forward pass process for each prediction is randomly sampled, thus its performance can be seen as the average performance of all  sampled architectures；
> - $\mathbf{E}$ denotes our final model without retraining, inheriting weights directly from the super-Transformer.
>
> Thus, we can conclude that the utilized evolutionary search effectively finds high-performance architectures to some extent.
>
>
> ## Response to Question 2
>
> In Figure 2, there are two main processes: **(1)** training a super-net and **(2)** searching via an evolutionary algorithm.  Here the first process will not be introduced because it has been presented in **Section 1-2 of Appendix**.
>
> The second process contains six steps,
> - First, i.e., **Population initialization** in Figure 2,  randomly generate $Pop$ individuals and form a population $P$;
> - Second, i.e., **Mating pool selection**,  randomly select two individuals from $P$ each time and keep the better one in $P'$ until the size of $P'$ equal to $P$;
> - Third, i.e., **Offspring generation**, randomly select two individuals from $P'$ each time,  apply crossover and mutation to the two individuals to generate two offspring individuals, forming offspring $Q$;
> - Fourth, i.e., **Individual evaluation**,  decode each individual in $Q$ to get its neural architecture, and directly evaluate each architecture's performance based on the inherited weights from the supernet;
> - Fifth, i.e., **Search space reduction**, (1) **Determine which operations are available**: As you can see, each column of figures in Figure 2 presents the available operations' encoding; the first column $[1,2,3,4]$ represents the available operations' encoding for the first bit is $[1,2,3,4]$, where $0$ is unavailable because it has been deleted. (2) **Compute the score of each available operation**.  (3) **Delete the operations whose scores are unpromising**: In Figure 2, X upon two figures represents these two operations are deleted in current generation.
> - Sixth, **Individual selection**, keep better individuals in $P$ and $Q$ to form the next population.
>
> ## Response to Question 3
> For the first concern, we would like to clarify that the increased computational cost is a trade-off we made to achieve improved performance and explore the potential of the super-Transformer. While it does come with a higher cost, it allows us to capture more complex student-exercise interactions, leading to better results.
>
> To answer the second concern,  a variant **ENAS-KT(f)** is created: the embedding size $D$ is halved to 64 for the supernet, the number of training epochs is also halved to 30, but the best-found architecture holds the same settings as **ours** for a fair comparison.
>
> As shown in **Table II** of   ***global_response***,  **Searched** holds a 0.8036 AUC value. Although its performance is worse than **ours**, as shown in **Fig.1(a)**, its overall cost only takes **9.1 hours** (**3.8hours** for supernet training, **1.7 hours** for evolutionary search, and 3.6 hours for final training), which is fewer than **ours** and competitive to *NAS-Cell* and others.
>
> In summary, current settings adopted in the proposed approach provide significantly better performance with a much higher cost, but our approach can also achieve competitive results at a much lower cost, whose performance is still better than other KT approaches and whose cost is competitive.
>
> ### We appreciate your valuable feedback and will revise the paper based on the above.

---

### Official Review · Reviewer_rdQD · 2023-07-07

**Soundness:** 2 fair
**Presentation:** 3 good
**Contribution:** 2 fair
**Rating:** 6
**Confidence:** 3

**Summary:**

The architecture of a transformer model for a knowledge tracing task is searched for using evolutionary computation. The found solutions perform well.



**Strengths:**

- ANN design is difficult
- KT is important
- Simple NAS method

**Weaknesses:**

- Interpretability of the solution, i.e. ANN
- Limited discussion regarding computational cost. E.g. how much improvement for the additional search
- Readability of only acronyms introduced, e.g. LFA, PFA, KTM
- Figure 1 is non-intuitive
- Limited discussion regarding transfer of the solutions to other KT datasets

**Questions:**

Questions:
- What is "Fusion" method?
- l135 Mashed=Masked?
- Is a supernet with EA a fancy dropout?
- What is the magnitude of the search space? Is it N^2 * C^{2N} ?
- What post-hoc adjustment did you use?
- Were the ablation results significant?


**Limitations:**

 none.

---

> ### Author Rebuttal · Authors · 2023-08-07
>
> ## Response to Question 1
> *The 'fusion' method* refers to that **a hierarchical fusion module** is employed in our approach to **aggregate** the input features fully.
>
> In **the hierarchical fusion module**,  the pairwise concatenation between input features is first performed to obtain some intermediate temporary features, then aggregate all temporary features as the final input feature by concatenation.
>
> The above **aggregation** process for input features is called *the 'fusion' method*.
>
> ## Response to Question 2
> We apologize for ***the typo in line 135***.  It should be $Masked\\_MHSA(\cdot)$: a masked multi-head self-attention module in Eq.(3) of the paper. The mask operation prevents the current position from accessing the later positions. Many thanks and we will revise this error.
>
> ## Response to Question 3
> In my opinion, the answer is yes. In a network with dropout, some weights will be set to active while others are set to inactive (zeros) to get the output of the network, the output of the network is only determined by the activated weights. Actually, in the proposed supernet, there exist many paths from its input to its output, and each path represents a sub-model. During the search with EA, when one path is active, other paths are set to inactive in the supernet, and the output of the supernet is only determined by the activated sub-model, i.e. the sub-model's output is the output of the supernet. Here the EA is used to find the best path in the supernet.
>
> As can be seen, the search process of the supernet with EA is similar to dropout, where some paths are set to inactive to get the model output. Thus we agree with the reviewer that the supernet with EA is dropout-like.
>
> ## Response to Question 4
> The magnitude of the search space is  $Num^2*2^{2(Num-1)}\*(3\*3\*5)^{2N}$, where $Num$ denotes the number of available input features and $N$ represents the number of encoder/decoder blocks.
>
> To be specifically, the architecture in the search space can be represented by $ [\mathbf{b_{En}},\mathbf{b_{De}}, 2N \times (lo\in\\{0,1,2,3,4\\}, go_1\in\\{0,1,2\\}, go_2\in\\{0,1,2\\})]$. $\mathbf{b_{En}}$ ($\mathbf{b_{De}}$)  is a binary vector with the length of $N$, whose at least one bit is 1.
>
> For $\mathbf{b_{En}}$ in the feature selection part, it is first to determine which bit is 1, leading to $Num$ choices;
> and the remaining $Num-1$ bits will provides $2^{Num-1}$ choices; as a results, there are $Num\*2^{Num-1}$ choices for  $\mathbf{b_{En}}$.
>
> It is same for $\mathbf{b_{De}}$ to know there are also $Num\*2^{Num-1}$ choices.
>
> For each encoder/decoder block, $lo$ has 5 candidate operations, $go_1$ has 3 candidate operations, and  $go_2$ also has 3 candidate operations, which leads to $5\*3\*3$ choices in each block; thus, $N$ encoder blocks and $N$ decoder blocks provides $(5\*3\*3)^{2N}$ choices totally.
>
> In summary, $\mathbf{b_{En}}$, $\mathbf{b_{De}}$ and $2N$ blocks totally provide $Num\*2^{Num-1}\times Num\*2^{Num-1}\times(5\*3\*3)^{2N}$ choices, i.e., $Num^2*2^{2(Num-1)}\*(3\*3\*5)^{2N}$.
>
> ## Response to Question 5
> We did not use any post-hoc adjustment method for the Wilcoxon rank sum test, because we compared each KT approach with our approach to assess whether there is a significant difference.
>
> The Wilcoxon rank sum test, also known as the Mann-Whitney U test, is typically used to compare two independent groups. In this context, post-hoc tests are not necessary because we're only comparing two groups.  Post-hoc tests are typically used in the context of analysis of variance (ANOVA) to compare multiple groups after finding a significant difference.
>
> In this paper, we employed the Wilcoxon rank sum test to analyze the significance because this test method has been used for various scenarios, such as the performance comparison in evolutionary computation [1~3].
> In these studies, the Wilcoxon rank sum test is directly used to assess whether there is a significant difference between one compared approach and their approach without using additional post-hoc tests.
>
> [1] Song Z, et al. Balancing Objective Optimization and Constraint Satisfaction in Expensive Constrained Evolutionary Multi-Objective Optimization[J]. IEEE Transactions on Evolutionary Computation, 2023.
>
> [2] Cheng R, et al. A reference vector guided evolutionary algorithm for many-objective optimization[J]. IEEE Transactions on Evolutionary Computation, 2016.
>
> [3] Kropp I, et al. Improved Evolutionary Operators for Sparse Large-Scale Multiobjective Optimization Problems[J]. IEEE Transactions on Evolutionary Computation, 2023.
>
> ## Response to Question 6
> To answer the question, we have executed the significance analysis based on the previous ablation results,
> and the overall results have been summarized in  **Table II** of ***global_response***.
>
> As can be seen, the Wilcoxon rank sum test with a significance level α=0.05 is performed to compare each variant architecture
> with **ours**. Finally, we can find that **ours** is significantly better than each variant architecture, demonstrating that the ablation results are significant.
>
> ## Response to Weakness
> 1.The found solution indeed does not have good interpretability, which will be one of our future research directions.
>
> 2.**Fig.1(a)  and Table II** show its cost and improvement: the supernet after training holds a 0.7847 AUC value, and the search process takes 9.7 hours to find an architecture $\mathbf{E}$ (**ours**  without retraining) having a 0.7969 AUC. The high improvement demonstrates that the additional search is deserved.
>
> 3&4.We will give all full names for all acronyms, and replot Figure 1 for better understanding and intuitive observation.
>
> 5.We ever considered a similar question, but the solutions cannot be directly used for other datasets because the available input features of other datasets are not the same as the two datasets in our approach.
>
> ### We appreciate your valuable feedback and will revise the paper based on the above.

---

> > ### Comment · Reviewer_rdQD · 2023-08-17
> >
> > These responses satisfy my questions and reinforce my rating.

---

> > > ### Author Response · Authors · 2023-08-17
> > > **Many thanks for your feedback.**
> > >
> > > We are grateful for taking your valuable time to review our paper and point out some issues.

---

### Author Rebuttal · Authors · 2023-08-04

We sincerely appreciate the valuable feedback provided by all the reviewers. We have carefully addressed their questions and concerns in our response, aiming to provide satisfactory answers.

Here we uploaded a file named "global_response" to show some necessary results and comparisons, which contains two tables and one figure.

 In all subsequent responses to reviewers,  this file is termed ***global_response*** for easy presentation and convenient discussion.

---

### Decision · Program_Chairs · 2023-09-21

**Decision:**

Accept (poster)

**Comment:**

The paper studies the problem of neural architecture search for knowledge tracing in educational applications. The goal is to find an optimized transformer architecture that can better select the input features and balance different modeling aspects, including the student's forgetting behavior. The proposed method, ENAS-KT, is based on designing a novel search space and an evolutionary search algorithm suitable for knowledge tracing. Extensive experimental evaluation on two large-scale benchmarks showcases the effectiveness of the proposed method. The reviewers acknowledged that the paper considers an important problem setting and that the proposed neural architecture search method is of practical interest. However, the reviewers also raised several concerns and questions in their initial reviews. We thank the authors for their detailed responses and for actively engaging with the reviewers during the discussion phase. The reviewers appreciated the responses, which helped in answering their key questions. The reviewers have an overall positive assessment of the paper, and there is a consensus for acceptance. The reviewers have provided detailed feedback in their reviews, and we strongly encourage the authors to incorporate this feedback when preparing the final version of the paper.